# The essential clathrin adapter protein complex-2 is tumor suppressive specifically in vivo

Seth P. Zimmerman[1], Lili B. DeGraw[1] & Christopher M. Counter ®[1,2] ✉

The microenvironment is a rich source of new cancer targets. We thus used a targeted single-guide RNA library to screen a panel of human pancreatic cancer lines for genes uniquely affecting tumorigenesis. Here we show inactivation of the Adapter Protein complex-2 of clathrin-mediated endocytosis reduces cell growth in vitro, but completely oppositely, promotes tumor growth in vivo. In culture, loss of the complex reduces transferrin endocytosis and iron import required for cell fitness. In tumors, alternative iron transport pathways allow pro-tumor effects of Adapter Protein complex-2 loss to manifest. In the most sensitive case, this is attributed to reprogramming the plasma membrane proteome, retaining integrins on the surface leading to Focal Adhesion Kinase phosphorylation and induction of proliferative signals. Adapter Protein complex-2 function in tumorigenesis is thus dependent upon the microenvironment, behaving as a common essential gene in culture via iron import, but as a tumor suppressor in tumors via integrin trafficking.

The tumor microenvironment (TME) plays a crucial role in tumorigenesis and therapeutic outcomes[1–3]. Highly lethal Pancreatic Ductal Adenocarcinoma (PDAC)[4] exemplifies this, as the cancer cells are embedded in a dense fibrotic extracellular matrix (ECM)[2]. Phosphatidylinositol (PtdIns) metabolism regulates signaling and membrane trafficking, crucial for tumor/TME interactions[5–9], is enriched in druggable proteins[5,6], contains a known proto-oncogene (PIK3CA) and tumor suppressor (*PTEN*)[5,6], and the PI3K/AKT pathway is activated in PDAC[10], with additional PtdIns kinases implicated in this cancer[11,12]. However, the role of PtdIns metabolism in plasma membrane trafficking has not been studied systematically in PDAC.

One mechanism by which PtdIns metabolism regulates plasma membrane trafficking is through the Adapter Protein complex-2 (AP2). AP2 binds membrane protein cargos and PI(4,5)P$_2$, recruiting PIP5K1C, thereby increasing the local PI(4,5)P$_2$ concentration, clathrin assembly, and cargo internalization[13,14] as a way of regulating receptors and their activity[15–17]. The transferrin receptor[18,19] and integrins[20,21] are well-established AP2 cargos that mediates iron import[18,19] and play a critical role in tumor/ECM interactions[22,23], respectively.

As the TME is an in vivo phenomenon, we sought to interrogate PtdIns metabolism in tumors. Indeed, CRISPR/Cas9-gene inactivation screens of whole-genome, single-guide RNA (*sgRNA*) libraries have identified genes critical for growth of tumor cell lines in culture but exclude the important TME context. Owing to their large size[24,25], these libraries have proven difficult to use in vivo. Nevertheless, in vivo screens have recently identified genes affecting tumor growth[25].

Here, we utilize a targeted *sgRNA* library to query whether PtdIns genes have a microenvironment-specific role in PDAC tumorigenesis. Specifically, we sought to identify genes that behave oppositely when inactivated in culture versus tumors, implying differential regulation by two different environments. Using this approach, we identify AP2 as common essential in vitro but tumor suppressive in vivo, and deconvolute the involvement of AP2 in iron and integrin signaling underlying these two opposing phenotypes.

## Results

### AP2 *sgRNA*s are oppositely enriched in vitro versus in vivo

To screen for genes that behave in a opposite manner in vitro versus in vivo the five transcriptionally distinct human PDAC cell lines (BXPC-

[1]Department of Pharmacology & Cancer Biology, Duke University Medical Center, Durham, NC, USA. [2]Department of Radiation Oncology, Duke University Medical Center, Durham, NC, USA. ✉e-mail: count004@mc.duke.edu

3, PANC-1, MIAPACA-2, CFPAC-1, and HPAF-II; Fig. S1a) were each independently infected in quadruplicate with a lentiviral *sgRNA* library comprised of five *sgRNA*s targeting each of 112 genes implicated in Ptdlns metabolism, as well a positive- and negative control *sgRNA*s (Fig. S1b and Supplemental Data 1)[5,6] and grown in culture or as xenografts (Fig. 1a). Genomic DNA was isolated from the 20 initial cultures, the 20 cultures after two weeks of passaging, and the 40 derived tumors, barcoded, and sequenced. Bioinformatic analysis confirmed the library complexity was maintained in all samples (Fig. S1c and Supplemental Data 2). *sgRNA*s targeting *PTEN* (*sgPTEN*) as well as *AP2A1* (*sgAP2A1*), *AP2A2* (*sgAP2A2*), *AP2B1* (*sgAP2B1*), *AP2M1* (*sgAP2M1*), and *AP2S1* (*sgAP2S1*) encoding the two α large subunits, the β2 large subunit, and the μ2 and σ2 small subunits of the AP2 complex were negatively enriched in four or all five of these cell lines when cultured. Consistent with PTEN being a tumor suppressor[26], *sgPTEN*s were positively enriched in BXPC-3, PANC-1, and MIAPACA-2 tumors, and weaklynegatively or not enriched in the other tumors, perhaps reflecting a redundancy in PI3K signaling[10]. Similarly, *sgAP2M1* and *sgAP2S1* were positively enriched in BXPC-3, PANC-1, and MIAPACA-2 tumors. *AP2A1/2* encode paralogues and *AP2B1* can be substituted by the corresponding AP1 subunit[27]. These redundancies are consistent with the variability that *sgAP2A1, sgAP2A2*, and *sgAP2B1* encoding the other subunits of AP2 were positively enriched in tumors (Fig. 1b, c and Supplemental Data 3). Interrogating a previous whole-genome *sgRNA* screen of lung cancer cell lines in 2-dimensional (2D) versus 3-dimensional (3D) culture[28] validated the differential enrichment of *sgPTEN, sgAP2M1*, and *sgAP2S1* (Fig. S1d). *sgAP2S1* was also positively enriched (Fig. S1e) in an in vivo, whole-genome screen of a murine lung cancer cell line[25]. Repeating the entire screen again in BXPC-3 and MIAPACA-2 cell lines validated the opposing enrichment of *sgPTEN, sgAP2M1*, and *sgAP2S1* in vitro versus in vivo (Fig. S1f). Finally, we note that *sgAP2*s were the most negatively-enriched in tumors in the ascending order of CFPAC-1/HPAF-II < < PANC-1/MIAPACA-2 < BXPC-3 (Fig. 1b), which we capitalize upon for comparisons (*see* below). Given that *sgRNA*s targeting the only two unique subunits of AP2 behaved in a similar opposing fashion in vitro versus in vivo in an unbiased screen, the progressively greater effect of *sgAP2* on promoting tumor growth in a panel of five human PDAC lines, and the connection of AP2 with tumorigenesis, we focused on AP2.

## AP2 loss inhibits cell growth in vitro

To directly test the effect of AP2 loss on culture growth, we determined the effect of inactivated AP2 on the culture growth of CFPAC-1, PANC-1, and BXPC-3 cells, reflecting the aforementioned progressively increased effect of AP2 loss on tumorigenesis. To begin, BXPC-3 cells were infected with lentiviruses encoding *sgAP2M1*, *sgAP2S1*, *sgPTEN*, and control *sgRNA* (*sgCTRL*). As loss of AP2 is detrimental in culture, the lentiviruses also express a doxycycline-inducible Cas9, followed by a 2a peptide, and then Green Fluorescent Protein[29] so that targeted genes could be inducibly inactivated and resultant cells verified by green fluorescence (Fig. S2a). Immunoblot confirmed a gradual reduction of the targeted genes in BXPC-3 cells upon increasing doxycycline (Fig. 2a). Given this, CFPAC-1, PANC-1, and BXPC-3 cells were similarly infected with these four lentiviruses and treated with doxycycline, after which immunoblot confirmed appropriate loss of AP2M1, AP2S1, and PTEN proteins in all three cell lines. *sgAP2M1* and *sgAP2S1* also reduce expression of the other subunit (Figs. 2b, c and S2b), consistent with loss of any one AP2 subunit destabilizing the obligate heterotetramer[30]. These cells were again treated with or without doxycycline in triplicate, and cell growth determined seven days later via crystal violet staining. In agreement with the genetic screen, conditional *AP2M1, AP2S1*, and *PTEN* inactivation variably, but significantly, decreased cell number in all three cell lines (Fig. 2d-i). Finally, four independent cultures of BXPC-3 cells encoding doxycycline-inducible *sgAP2S1*-GFP or *sgAP2S1*-mCherry and

*sgRNA*-resistant *AP2S1* were treated with or without doxycycline in triplicate, *sgRNA* expression was verified via fluorescence, after which cells were cultured for seven days and stained with crystal violet. This revealed that the loss of cell growth upon *AP2S1* inactivation was rescued by co-expressing *sgRNA*-resistant *AP2S1*, thereby discounting any other effect (Fig. S2c, d). Thus, disrupting AP2 function impairs culture growth.

## AP2 loss promotes tumor growth in vivo

To directly test the effect of AP2 loss on tumorigenesis, BXPC-3 cells expressing doxycycline-inducible *sgAP2M1, sgAP2S1, sgPTEN* as a positive control, or *sgCTRL* as a negative control were treated with or without doxycycline, cultured, and then injected into five mice per group, after which tumor size was regularly measured and mean tumor volume plotted versus time. Again, inducible gene inactivation was required owing to the essentiality of these genes in vitro. *sgAP2M1* and *sgAP2S1* significantly increased tumor growth compared to uninduced control cells, with *sgAP2S1* nearly as effective as inactivating the control tumor suppressor PTEN (Fig. 2j). To rule out tumor promotion due to doxycycline treatment, cells without an *sgRNA* that were treated with doxycycline were similarly tested, revealing that they did not exhibit enhanced tumor growth. We also confirmed the enhanced tumor growth upon AP2 loss with a second *sgAP2S1*, thereby independently validating the phenotype (Fig. S2e). Tide analysis[31] also found more insertions and deletions in the *AP2S1* gene in tumors compared to culture, suggesting a competitive advantage for *AP2S1* loss in vivo (Fig. S2f). Finally, we repeated the tumor analysis with PANC-1 and CFPAC-1 cells encoding doxycycline-inducible *sgAP2S1* versus *sgCTRL*, which revealed that *sgAP2S1* increased tumor growth of PANC-1 (Fig. 2k) but not CFPAC-1(Fig. 2l) cells compared to *sgCTRL*, consistent with the aforementioned cell line-specific gradient effect of AP2 loss on *sgRNA* enrichment in vivo (Fig. 1b).

To ascertain whether enhanced tumorigenesis upon disrupting AP2 is cell autonomous, we assessed the impact of AP2 loss on tumor spheroid growth in 3D culture. Due to inefficient inactivation of the essential AP2 genes by Cas9 leading to overgrowth of wild-type cells in culture, we turned to doxycycline-inducible *AP2M1 shRNA* (*shAP2M1*) to conditionally reduce AP2M1 expression in more cells in comparison to a control *shRNA* (*shCTRL*). Inducing *shAP2M1* in BXPC-3 cells reduced AP2M1 protein (Fig. 2m) and significantly increased tumor spheroid size compared to *shCTRL* (Fig. 2n) in three independent experiments (Fig. 2o). To independently validated these findings, lentiviruses encoding doxycycline-inducible GFP or Red Fluorescent Protein (RFP) with *shCTRL* or *shAP2M1* were used to generate stable GFP-*shCTRL*, RFP-*shCTRL*, and RFP-sh*AP2M1* BXPC-3 cells for growth competition assays. Mixing BXPC-3 cells expressing RFP-*shCTRL* or RFP-sh*AP2M1* with an equal number of GFP-*shCTRL* cells in two separate experiments revealed upon doxycycline treatment a reduction of RFP-sh*AP2M1* cells in 2D but an increase in 3D culture over time compared to the GFP-*shCTRL* cells (Fig. 2p, q). In contrast, induction of *shAP2M1* in CFPAC-1 cells depleted AP2M1 protein (Fig. 2r) but did not affect 3D culture growth (Fig. 2s) in this growth competition assay. These findings support that AP2 loss promotes tumor growth in a cell-autonomous fashion dictated by the microenvironment and cell type.

## AP2 loss inhibits cell growth in vitro via reduced iron import

As AP2 targets specific plasma membrane proteins for endocytosis[18,30,32–35], its cargos may underly the divergent phenotypes observed upon AP2 loss in vitro versus in vivo. In vitro, DepMap[36,37] defines *AP2M1* and *AP2S1* as common-essential genes based on negative enrichment of *sh/sgAP2M1/S1* in over 1800 whole-genome screens (Fig. S3a, b). These two genes are also the top co-dependencies for each other (Fig. 3a), namely *sh/sgRNA*s against both genes exhibit similar negative-enrichment patterns across cell lines[36,37]. In terms of a potential cargo, Transferrin Receptor 1 (TFR1 encoded by the *TFRC*

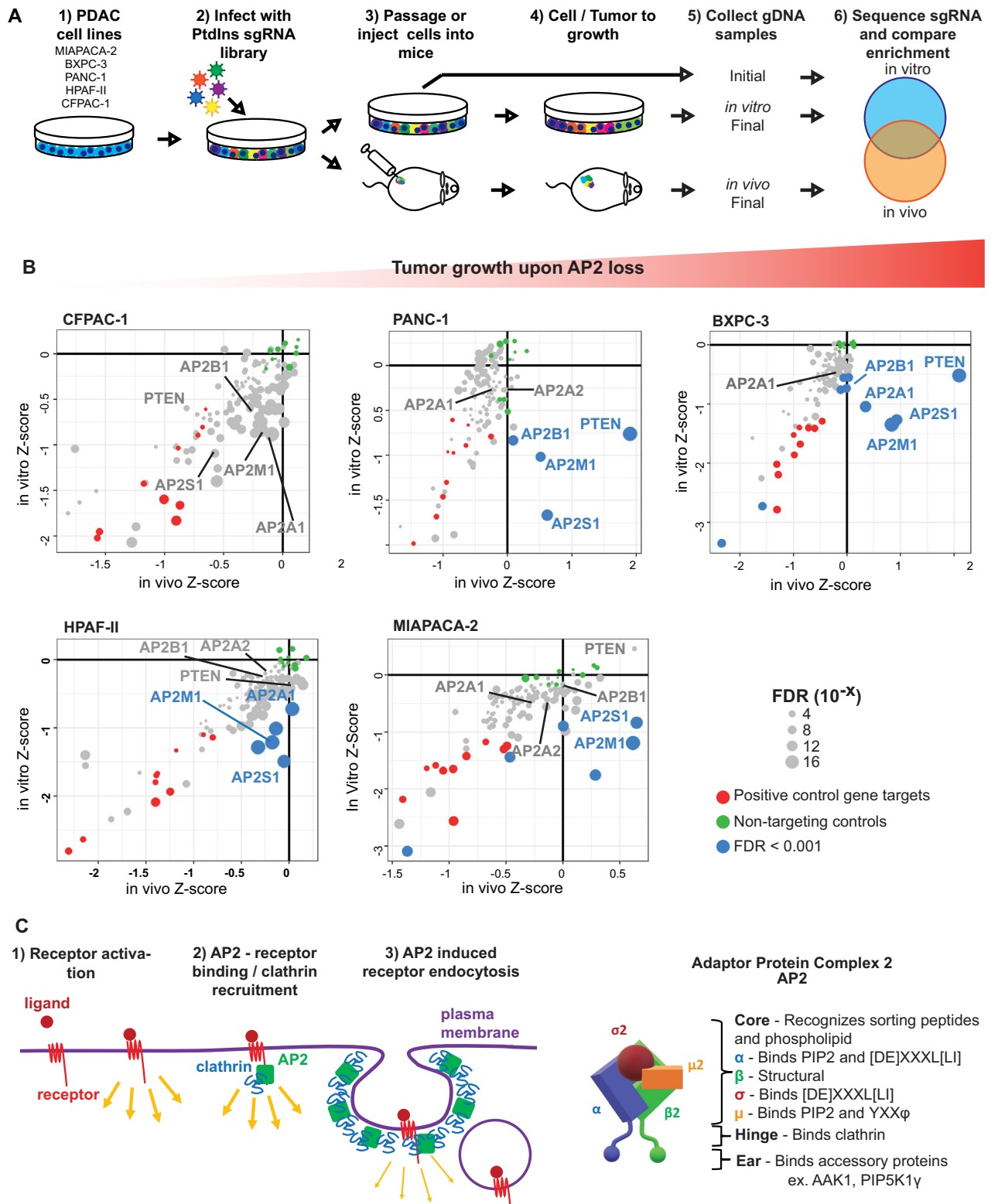

**Fig. 1 | PtdIns metabolism-targeted CRISPR/Cas9 screen identifies AP2 complex as a PDAC tumor suppressor in vivo and essential in vitro. A** Schematic of CRISPR/Cas9 loss-of-function screen. **B** A scatter-plot of in vitro (culture) and in vivo (xenograft) Z-scores representing loss-of-function gene effects from the screen. Cell lines ranked from left to right by in vivo enhanced tumor growth upon AP2 loss. False Discovery Rates (FDRs), represented by circle size, refer to comparison of normalized in vivo to in vitro Z-Scores across tested cell lines (for each cell line, in vivo sample $n = 8$ xenografts per condition, in vitro sample $n = 4$ technical replicates per condition; each sample was independently infected with library and cultured). **C** Schematic of AP2 complex function in endocytosis of membrane proteins and an illustrated representation of its structure and function. Source data are provided as a Source Data file. Illustrations adapted from refs. 14,32.

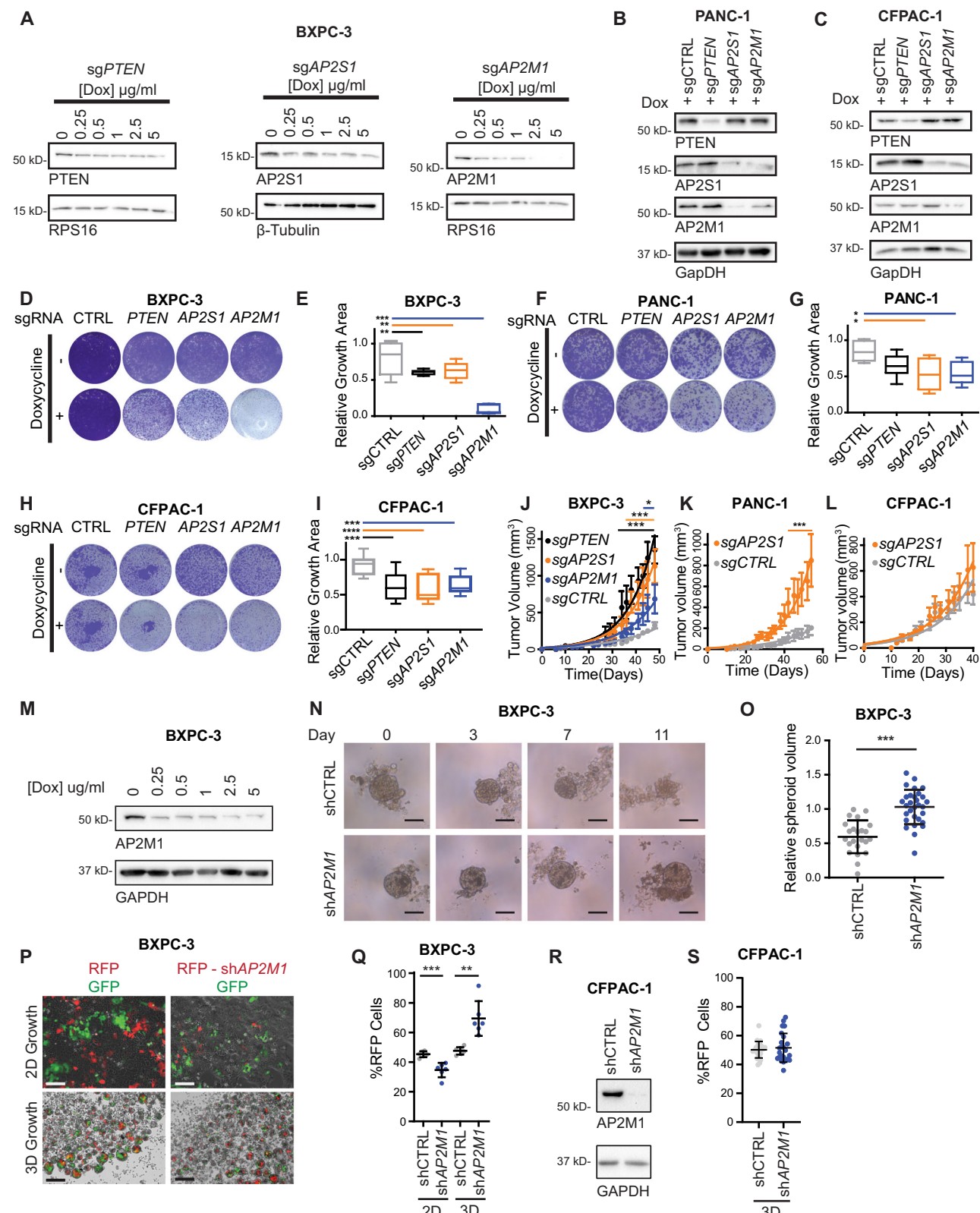

gene) bound to iron-loaded Transferrin (holo-TF) is imported via AP2-dependent endocytosis and is a major source of cellular iron[38]. *TFR1* is also common essential (Fig. S3c) and shares a significant co-dependency with both *AP2M1* and *AP2S1* (Fig. 3a), suggesting a functional connection. To therefore address whether TFR1 is the cargo underlying the reduced proliferation upon AP2 loss in cultured cells, BXPC-3 and CFPAC-1 cells, both of which grow poorly in vitro but only

the former grows better in vivo upon AP2 loss, expressing doxycycline-inducible *shAP2M1* or *shCTRL* transcriptionally linked to RFP to identify cells expressing the *shRNA*, were cultured in the presence of doxycy-cline and then incubated with TF-488, holo-TF conjugated to the green fluorescent dye Alexa Fluor 488. Cells were washed in a high salt and low pH buffer to remove extracellular fluorescence, fixed and assessed by fluorescence microscopy. As expected[18], microscopic analysis

**Fig. 2 | Direct test of AP2 subunit loss-of-function validates environment-dependent growth phenotype.** Representative immunoblots of the indicated proteins from (**A**) BXPC-3 (representative of *n* = 3 independent experiments), (**B**) PANC-1 (representative of *n* = 3 independent experiments), and (**C**) CFPAC-1 (representative of *n* = 3 independent experiments) cells with (**A**) dose-dependent or (**B**, **C**) single dose of doxycycline (Dox) induced expression of Cas9.
**D, F, H** Representative crystal violet stained (**D**) BXPC-3, (**F**) PANC-1, and (**H**) CFPAC-1 cell culture wells from dox-induced expression of Cas9. **E, G, I** Box and whisker plot of the median, IQR and min/max values from quantification of the relative growth area for (**D**, **F**, and **H**). E - *sgCTRL* and *sgAP2S1 n* = 12 wells total over 4 independent experiments; *sgPTEN* and *sgAP2M1 n* = 9 wells total over 3 independent experiments. G – *n* = 6 wells total over 2 independent experiments. I - 12 wells total over 4 independent experiments, 1-way ANOVA). **J–L** Plots of tumor volume versus time in days for tumors derived from (**J**) BXPC-3, (**K**) PANC-1, and (**L**) CFPAC-1 cells transduced with the indicated *sgRNAs* and induced with doxycycline compared to all uninduced control tumors (mean +/- standard error, J - *n* = 5 mice per *sgRNA* condition, *n* = 10 for control condition. **K** and **L** – *n* = 5 mice per condition. 2-way ANOVA). **M** Representative immunoblot of the doxycycline induction of

*shAP2M1* in BXPC-3 cells (representative of *n* = 4 independent experiments comparing 1 and 0 μg/ml doxycycline). **N** Representative phase contrast images of BXPC-3 spheroid growth assays over time (bar = 100 μm) (**O**) Scatter plot superimposed on mean and standard deviation of relative spheroid area quantification at day 11 compared to day 0 in N (*shCTRL* – *n* = 23 spheroids total over 5 independent experiments, *shAP2M1 n* = 30 spheroids total over 5 independent experiments; two-sided *t*-test). **P** Representative merged fluorescent and phase contrast micrographs of BXPC-3 competitive growth assays in 2D and 3D culture after 7 days of growth. **Q** Scatter plot superimposed on mean and standard deviation of % RFP cells after 7 days of growth, quantified from competitive growth assays in P (*n* = 6 technical replicates total over 2 independent experiments; two-sided *t*-test).
**R** Representative immunoblot of the doxycycline dose-response induction of *shAP2M1* in CFPAC-1 cells (representative of *n* = 3 independent experiments).
**S** Scatter plot superimposed on mean and standard deviation of % RFP cells after 7 days of CFPAC-1 spheroid growth competition (*n* = 12 technical replicates total over 2 independent experiments; two-sided *t*-test). *\*p* < 0.05, *\*\*p* < 0.01, *\*\*\*p* < 0.001, and *\*\*\*\*p* < 0.0001. Source data are provided as a Source Data file.

revealed reduced green fluorescence in BXPC-3 and CFPAC-1 cells with efficient knockdown of *AP2M1* (high RFP) compared to poor (low RFP) or no (*shCTRL*) knockdown in two independent experiments, reaching the same level achieved by blocking clathrin-mediated endocytosis with the clathrin inhibitor Pitstop2 (Fig. 3b–e). Thus, AP2 loss reduces the import of holo-TF.

We then tested whether this block in endocytosis reduced iron import and consequently culture growth. BXPC-3 and CFPAC-1 cells expressing doxycycline-inducible *shAP2M1* or *shCTRL* were again cultured the absence or presence of doxycycline, then increasing levels of iron supplementation in the form of Ammonium Ferric Citrate (AFC)[39] for seven days and then stained with crystal violet. In the absence of doxycycline, AFC gradually inhibited cell growth, as previously reported[40]. On the other hand, the potent inhibition of culture growth due to *shAP2M1* was gradually rescued with increasing concentrations of AFC, until high AFC levels expectedly[40] reduced cell fitness, a reproducible phenotype observed in quadruplicate samples tested independently twice (Fig. 3f-i). Thus, exogenous iron rescues the negative effect of AP2 loss in vitro.

We next tested whether the loss of AP2 depletes cellular iron in a TF-dependent manner. BXPC-3 cells expressing doxycycline-inducible *shAP2M1* or *shCTRL* were cultured in doxycycline, serum starved, and treated with the iron chelator Deferoxamine[41] (DFO) for one hour to remove intra- and extra-cellular sources of iron. Cells were then maintained in DFO, holo-TF, or AFC for 16 h and mitochondrial iron content was microscopically assayed by the iron sensitive fluorescent dye Mitto-Ferro Green. Cells expressing *shAP2M1* and treated with holo-TF had significantly reduced mitochondrial fluorescent staining compared to *shCTRL* cells, while AFC- and DFO-treated cells showed no difference compared to *shRNA* conditions (Fig. 3j, k). Thus, AP2 loss reduces cellular iron by inhibiting TF-dependent, and not TF-independent iron import. Additionally, the dose-response curve generated with increasing DFO concentrations revealed that AP2 loss rendered cells less sensitive to iron chelation (Fig. S3d), yielding a significant difference in the area under the curve compared to *shCTRL* cells (Fig. S3e), consistent with AP2 loss depleting cellular iron such that cells are rendered less sensitive to further chelation. Taken together, these data are consistent with AP2 loss preventing TFR1/holo-TF endocytosis, thereby lowering intracellular iron levels and reducing culture growth.

### In 3D culture and tumors intracellular iron is not depleted in cells lacking AP2

Since the reduction of iron import accounted for the decrease in cell growth upon disrupting AP2 in cultured cells, we queried whether intracellular iron is altered in 3D culture by quantifying the relative

expression of iron biomarker proteins in BXPC-3 cells with and without AP2. TFRC and the Iron Responsive Element Binding protein 2 (IREB2) are upregulated in settings of low iron whereas the Ferratin Heavy Chain (FTH1) is upregulated in the presence of high iron[42]. BXPC-3 cells expressing doxycycline-inducible *shAP2M1* or *shCTRL* were cultured in 2D versus 3D in the presence of doxycycline. Immunoblot confirmed expected reduction of AP2M1 protein (Fig. S3f, g). In 2D culture, AP2M1 depletion coincided with a significant increase in TFRC, a non-significant increase in IREB2, and no change in FTH1 expression compared to *shCTRL* control cells, supporting a reduction in intracellular iron. However, FTH1, TFRC, and IREB2 expression was not different in *shAP2M1* versus *shCTRL* cells when placed in 3D culture, supporting that intracellular iron is not affected by AP2 loss. Comparing 2D to 3D conditions revealed a non-significant trend in the reduction of TFRC and IREB2 with a significant increase in FTH1, supporting an increase in intracellular iron in 3D culture. Given this, we next determined the levels of *SLC11A1*, *MCOLN1*, *SLC40A1*, *SLC22A17*, *SCARA5*, and *STEAP3*, which are involved in TFR1-independent iron-transport pathways[43–50], as well as *TFRC*, which as noted above is the transferrin receptor, by qRT-PCR in 2D versus 3D culture of BXPC-3 and CFPAC-1 cells. This revealed an increase in most of these transcripts in both cell lines in 3D culture (Fig. S3h, i). To validate these results in vivo, BXPC-3 tumors grown in the absence or presence of AP2 were immunoblotted to compare the levels of FTH1, TFRC, and IREB2. As expected, FTH1 and IREB2 levels were similar in vivo regardless of AP2 expression, supporting iron import being restored in vivo in the absence of AP2. We note however that TFRC levels were elevated in the AP2-deficent tumors (Fig. S3j, k). Since loss of AP2 directly alters the trafficking of TFRC, this increase may be the result of reduced endocytosis rather than increased synthesis. In summary, these data are consistent with the decrease observed in intracellular iron upon loss of AP2 in 2D culture, as assessed by multiple biomarkers, being overcome in 3D culture and in tumors.

### Integrins are retained on the plasma membrane in tumors exhibiting enhanced growth upon AP2 loss

We next sought to identify the pro-tumorigenic signal manifested by the loss of AP2 in vivo. As AP2 endocytoses specific protein cargoes[14], thereby altering tumor/TME interface, we characterized changes to the plasma membrane proteome upon AP2 loss in vivo. CFPAC-1, PANC-1, and BXPC-3 cells expressing doxycycline-inducible *sgAP2S1* were treated or not with doxycycline and injected into five mice each. The plasma membrane fraction was enriched through membrane protein isolation for each of the 30 resulting tumors, and associated proteins identified by mass spectrometry (Supplemental Data 4). This revealed a modest increase in the total number of proteins altered upon AP2

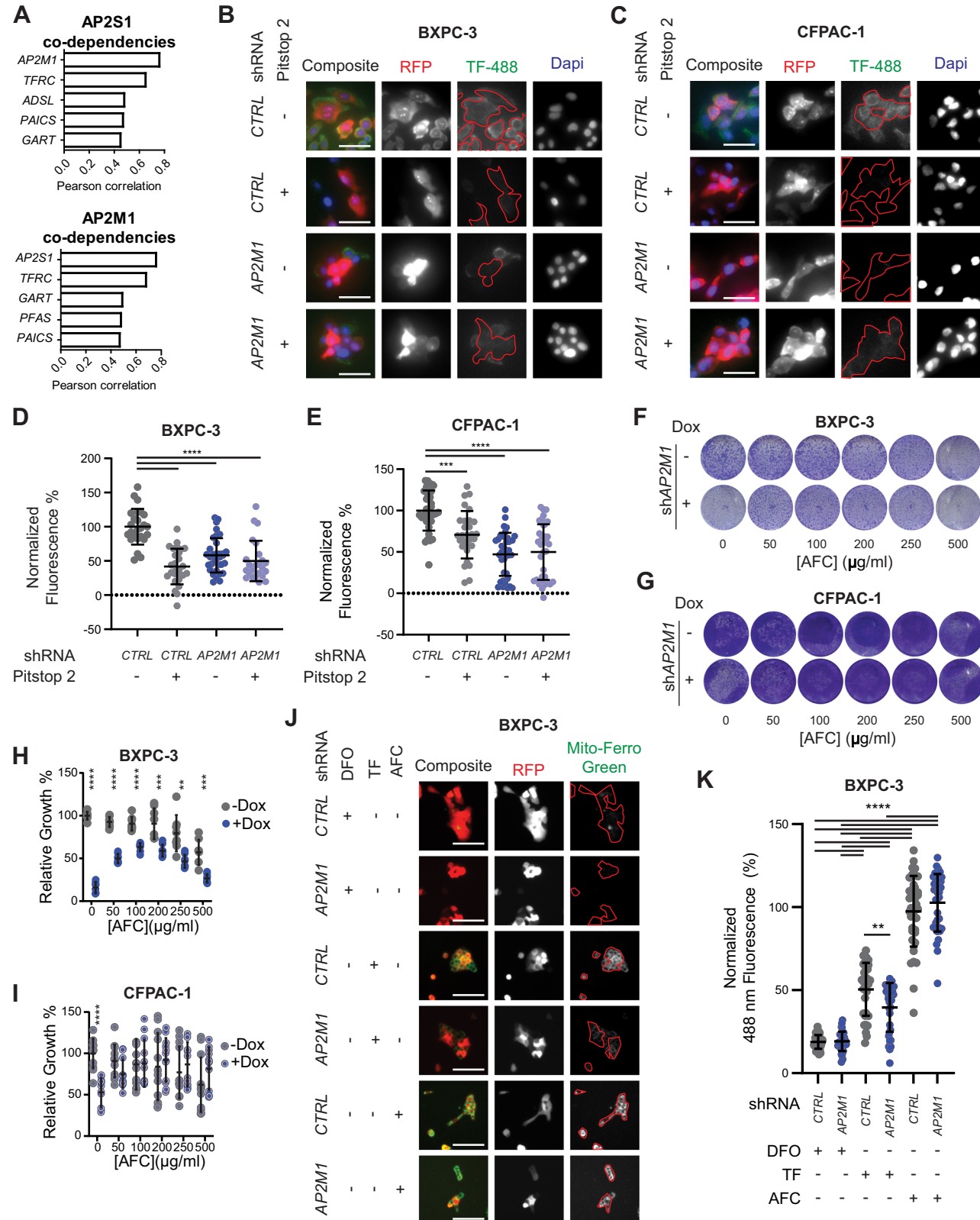

loss in tumors across the three lines (Fig. 4a). While there was limited overlap in enriched proteins between tumor types, Gene Ontology analysis revealed positive enrichment in cell adhesion-related terms, such as cell junctions, focal adhesion, and cell-substrate junctions in tumors derived from BXPC-3 and PANC-1 that are responsive to AP2 loss (Fig. 4b–d). Notably, eleven integrin subunits were prominently enriched in the plasma membrane proteome of BXPC-3 tumors and

one overlapping integrin subunit was enriched in PANC-1 tumors upon AP2 inactivation, suggesting a potential link to integrins.

Integrins are a family of transmembrane ECM-adhesion hetero-dimers comprised of 24 unique combinations of 18 α and 8 β subunits[16], which can undergo AP2-dependent endocytosis through direct and secondary interactions with AP2[20,21]. Once internalized, they are trafficked to the lysosome for degradation or recycled back to the

**Fig. 3 | AP2-dependent transferrin endocytosis supports cell culture growth through iron regulation. A** Plot of the Pearson Correlation coefficient for top DepMap[36,37] co-dependencies of *AP2M1* and *AP2S1*. **B, C** Representative fluorescent micrographs of internalized fluorescently labeled TF in (**B**) BXPC-3 and (**C**) CFPAC-1 cells. RFP correlates to the expression of the indicated *shRNA*. Red highlighted region indicates the image area quantified in D and E. Bar = 50 µm. **D, E** Scatter plot superimposed on mean and standard deviation of normalized TF-488 fluorescence in RFP-expressing cells quantified from B and C respectively (D − *n* = 25, 28, 35, and 25 total images respectively over 2 independent experiments, **E**−*n* = 36, 30, 40, and 35 total images respectively over 2 independent experiments; one-way ANOVA). **F, G** Representative crystal violet stained (**F**) BXPC-3 and (**G**) CFPAC-1 cell culture wells with doxycycline (Dox) induced expression of *shAP2M1*. Cells were treated with increasing concentrations of Ammonium Ferric Citrate (AFC). **H, I** Scatter plot superimposed on mean and standard deviation of relative growth quantified from (**F** and **G**) (**H**−*n* = 6 wells total per condition over 2 independent experiments. **I**−*n* = 9 wells total per condition over 3 independent experiments.; Multiple two-sided *t*-tests). **J**) Representative fluorescent micrographs of BXPC3 cells stained with Mito-Ferro Green dye, expressing the indicated *shRNAs* and treated with Deferoxamine (DFO), Transferrin (TF), or AFC. RFP correlates to the expression of the indicated *shRNA*. Red highlighted region indicates the image area quantified in (**K**). Bar = 50 µm. **K** Scatter plot superimposed on mean and standard deviation of Mito-Ferro green in RFP-positive cells represented in (**J**) (*n* = 31, 35, 35, 36,40, and 37 total images respectively over 2 independent experiments; one-way ANOVA). \**p* < 0.05, \*\**p* < 0.01, \*\*\**p* < 0.001 and \*\*\*\**p* < 0.0001. Source data are provided as a Source Data file.

plasma membrane to re-engage the ECM[16]. To validate the observed accumulation of integrins on the tumor cell surface, we focused on the most enriched integrins upon AP2 loss, namely ITGB1, ITGB6, ITGA2, and ITGA3. Immunoblot analysis of ten tumors derived from BXPC-3 cells with or without induction of *sgAP2S1* revealed that while the degree of AP2S1 expression varied somewhat between tumors and the total amount of these integrins differed, ITGB1, ITGB6, and ITGA2 were significantly enriched in the plasma-membrane fraction of tumors in which *sgAP2S1* was induced, with ITGA3 trending in the same direction (Figs. 4e, f and S4a–c). Given these results, we tested whether integrins were also retained on the plasma membrane in 3D culture. BXPC-3 cells expressing doxycycline-inducible *shAP2M1* or *shCTRL* and RFP to identify cells expressing the *shRNA* were cultured as tumor spheroids in the presence of doxycycline and ITGB6 at the plasma membrane detected by immunofluorescent staining with an anti-ITGB6/ITGAV antibody without permeabilizing the spheroids cell membrane. Microscopic analysis revealed an increase in ITGB6 at the plasma membrane upon AP2 loss in two independent experiments (Figs. S4d, e). We next investigated whether this retention reflected a defect in endocytosis. BXPC-3 and CFPAC-1 cells expressing doxycycline-inducible *shAP2M1* or *shCTRL* and RFP were cultured in the presence of doxycycline and in the presence or absence of Pitstop2. Cells were then incubated with anti-ITGB1-488, an anti-ITGB1 antibody conjugated to a 488 nm fluorescent dye. Cells were washed in a high salt and low pH buffer to remove extracellular fluorescence, fixed, and assessed by fluorescence microscopy. BXPC-3 cells expressing *shAP2M1* or treated with Pitstop2 showed an ~40% reduction in ITGB1 endocytosis compared to untreated cells expressing *shCTRL*. Additionally, treatment of *shAP2M1*-expressing cells with Pitstop2 showed no additional reduction in fluorescence signal compared to *shAP2M1*-expressing or Pitstop2-treated cells indicating that all clathrin-mediated endocytosis of ITGB1 is AP2-dependent in BXPC-3 cells (Fig. 4g, h). Interestingly, CFPAC-1 endocytosis of ITGB1 was not affected by AP2M1 depletion or Pitstop2 treatment (Fig. 4i, j), validating the results of the plasma membrane proteomics and suggesting that integrin endocytosis is independent of clathrin in cell lines where AP2 loss did not enhance tumor growth. Taken together, these data support reduced endocytosis of integrins specifically in cells that are more tumorigenic in the absence of AP2.

### FAK activation in tumors exhibiting enhanced growth upon AP2 loss
The interactions between integrin heterodimers and ECM not only function to physically adhere a cell to the ECM but also as a receptor-ligand interaction, transmitting a signal across the plasma membrane, which is enhanced by physical tension across the interaction[51]. Once activated, integrins transduce a proliferative signal into the cytoplasm through their interaction with *F*ocal *A*dhesion *K*inase (FAK), leading to autophosphorylation at tyrosine 397[52]. To explore whether integrin enrichment at the plasma membrane in response to AP2 loss leads to FAK activation, total and phosphorylated FAK (pFAK) as well as AP2S1

protein levels were determined by immunoblotting lysates from ten tumors derived from BXPC-3 cells expressing doxycycline-inducible *sgAP2S1*, half being treated with doxycycline and the other half not prior to injection. As expected, tumors without AP2S1 showed a significant increase in pFAK compared to those with AP2S1 (Fig. 4e, f). Given these results, we explored whether FAK is also phosphorylated in 3D culture upon AP2 loss. BXPC-3 cells expressing doxycycline-inducible *shAP2M1* or *shCTRL* were cultured independently twice in 3D in the absence or presence of doxycycline. Immunoblot revealed an increase pFAK when AP2M1 was depleted. Moreover, this increase was reversed by incubation with a FAK (FAKi; PF-573228) or integrin αvβ5, α5β1, αvβ3, and αIIbβ3[53] (cilengitide) inhibitor (Fig. 4k–m). These effects were recapitulated in cell growth competition. Namely, when equal number of BXPC-3 cells expressing RFP-*shAP2M1* or GFP-*shCTRL* were co-cultured in 3D, red-fluorescent cells were significantly enriched over green-fluorescent cells when compared to co-cultured GFP-*shCTRL* and RFP-*shCTRL* cells, indicating enhanced 3D growth upon AP2M1 depletion. Further, addition of cilengitide or FAKi repressed the enrichment of RFP-*shAP2M1* over GFP-*shCTRL* cells (Fig. 4n). Comparing pFAK levels in BXPC-3 cells grown in 2D versus 3D revealed that 2D growth produced exceptionally higher levels of pFAK (Fig. S4f). Similarly, CFPAC-1 cells grown in 2D exhibited saturated pFAK activity, while pFAK was undetectable in 3D culture, the latter being independent of AP2M1 depletion (Fig. S4g). Together, these data suggests that the loss of AP2 reduces endocytosis of integrins, and their accumulation of the plasma membrane leads to FAK activation and a proliferative signal. Conversely, in tumors that do not grow better upon AP2 loss, integrins are not internalized by clathrin-mediated endocytosis and correspondingly neither accumulate integrins on the cell surface nor activate FAK in the absence of AP2.

### Transcriptional reprogramming by AP2 loss in vitro versus in vivo
We propose that loss of AP2 blocks iron import in cultured cells, leading to a growth arrest, that is overcome by TF-independent iron import in tumors, which allows other effects of AP2 to manifest, namely reduced endocytosis of integrins and the corresponding proliferative signal specifically in cells whereby integrin import is clathrin-dependent. To assess the effect of these changes on the transcriptome, we performed total RNA sequencing on triplicate cultures and five independent tumors derived from CFPAC-1, PANC-1, and BXPC-3 cells, representative of the gradual increase in tumor growth upon AP2 loss, with and without *AP2S1* inactivation (Figs. 5a and S5a). Transcriptomes were normalized (Supplemental Data 5) and the top 100 most differentially enriched genes analyzed. Not surprising, given that these three cell lines were chosen for their diverse transcriptional profiles (Fig. S1a), samples clustered based on cell type, and then on whether derived from culture or tumors (Fig. S5b). Beyond this there were three transcriptional patterns.

First, transcripts of proteins involved in iron transport were significantly altered in tumors from all cell types when compared to

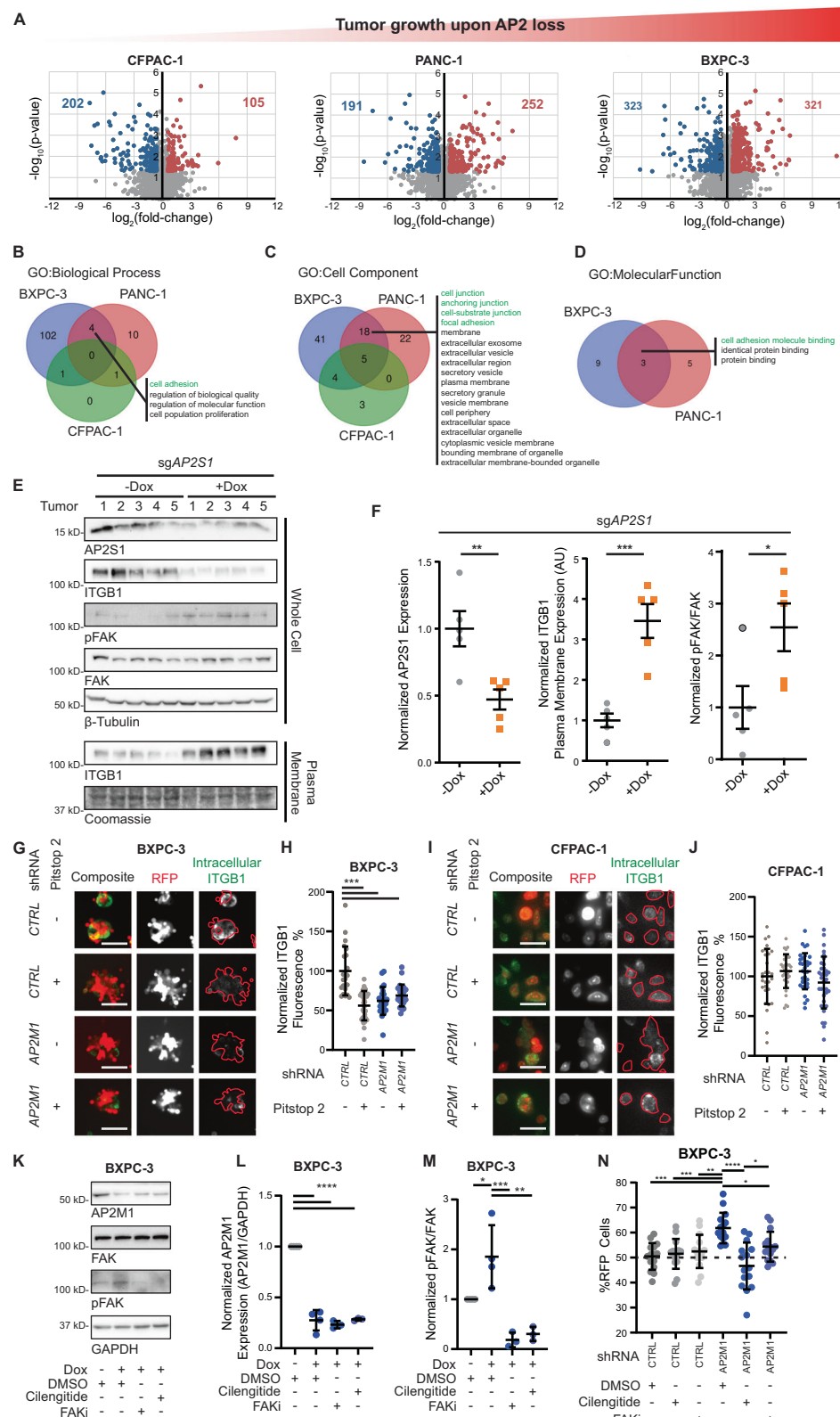

cultured cells. *TFRC* mRNA encoding TFR1 was higher in culture, while the genes *SLC11A1, MCOLN1, SLC4OA1, SLC22A17, SCARA5*, and *STEAP3* encoding proteins involved in TFR1-independent iron-transport pathways[43–50] were generally higher in the tumors (Figs. 5b, c and S5c). Furthermore, the levels of *IREB2* and *ACO1* mRNA, the later encoding <u>*aco*</u>nitase *1* (a.k.a. IREB1) that is also upregulated in iron-deplete conditions[42], were lower in tumors compared to cultured cells across

all three cell lines (Figs. 5b, c and S5d,e). These data taken together with previous observations support the reduction in iron import observed upon AP2 loss in culture is overcome in 3D culture and tumors by TFR1-independent iron import.

Second, transcriptional changes mirrored the effect of AP2 loss in tumors. Namely, CFPAC-1 tumors showed significant changes in only two genes, PANC-1 in 299 genes, and BXPC-3 in 423 genes (Fig. 5a).

**Fig. 4 | AP2 loss of function reprograms cell membrane proteome leading to increased Integrin and FAK pathway activity. A** Volcano plot of representing differential plasma membrane proteins comparing tumor samples from each cell type with and without *AP2S1* (red – positive enrichment in absence of AP2, blue – negative enrichment, *n* = 3 technical replicates per condition; two-sided heteroscedastic t-test; fold-change > 2 and *p* < 0.05). **B**, **C**, **D** Venn diagrams representing overlapping GO terms from the indicated class. GO analysis was performed on proteins positively enriched at the cell membrane with the *AP2S1* loss. Terms shared exclusively between PANC-1 and BXPC-3 cells are labeled and green highlighted terms are related to cell adhesion. **E** Immunoblot analysis of the indicated proteins from 10 tumors expressing *sgAP2S1* and treated or not with doxycycline (Dox) before implantation to induce Cas9 (representative of *n* = 1 immunoblot of 5 independent tumors per condition). **F** Scatter plot superimposed on mean and standard deviation of AP2S1, plasma membrane ITGB1, and phosphorylated FAK from E (*n* = 5 tumors per condition, two-sided *t*-test). **G**, **I** Representative fluorescent micrographs of internalized fluorescently labeled ITGB1 in (**G**) BXPC-3 and (**I**) CFPAC-1 cells. RFP correlates to the expression of the indicated *shRNA*. Red highlighted region indicates the image area quantified in H and J. Bar = 50 μm. **H**, **J** Scatter plot superimposed on mean and standard deviation of normalized ITGB1-488 fluorescence in RFP expressing cells quantified from G and I respectively (**H**) - *n* = 22, 29, 23, and 24 total images respectively over 2 independent experiments, J – *n* = 29, 33, 33, and 34 total images respectively over 2 independent experiments; one-way ANOVA). **K** Representative immunoblot of the indicated proteins from BXPC-3 cells cultured in 3D and expressing the indicated *sgRNA* and treated or not with doxycycline (Dox), Cilengitide, or FAK inhibitor (FAKi) (representative of *n* = 3 replicate immunoblots with inhibitor samples and 1 without over 4 independent experiments). **L** Scatter plot superimposed on mean and standard deviation of AP2M1 expression normalized to GAPDH from K (*n* = 4, 4, 3, and 3 independent experiments respectively, one-way ANOVA). **M** Scatter plot superimposed on mean and standard deviation of pFAK expression normalized to total FAK from K (*n* = 4, 4, 3, and 3 independent experiments respectively, one-way ANOVA). **N**) Scatter plot superimposed on mean and standard deviation of % RFP cells after 7 days of growth, quantified from BXPC-3 competitive growth assays treated or not with doxycycline (Dox), Cilengitide, or FAK inhibitor (FAKi). (*n* = 15 technical replicates per condition over 3 independent experiments; one-way *ANOVA*). *\*p* < 0.05, *\*\*p* < 0.01, *\*\*\*p* < 0.001, and *\*\*\*\*p* < 0.0001. Source data are provided as a Source Data file.

These data support AP2 loss progressively altering cellular signaling in conjunction with enhanced tumorigenesis.

Third, cultures exhibited transcriptional hallmarks associated with tumorigenesis were diminished in tumors but restored upon AP2 loss. Namely, we performed Gene Set Enrichment Analysis (GSEA) of the culture and tumor transcriptome dataset using 50 curated "hallmark" gene sets representing well-defined biological states[54]. This revealed upregulation of KRAS signaling, TGFβ signaling, PI3K_AKT_MTOR signaling, MTORC1 signaling, MYC targets, and E2F targets hallmark gene sets in culture versus tumors and in tumors with versus without *sgAP2S1* induction. These signatures were more highly enriched in BXPC-3 compared to PANC-1 datasets (Figs. 5b and S5c, e), consistent with the more robust tumor growth of BXPC-3 cells in the absence of AP2, and are known to be downstream of activated FAK[51,52,55,56]. These data support activated FAK stimulating a proliferative transcriptional response.

Taken together, these three transcriptional patterns suggest that the loss of AP2 in culture reduces endocytosis of TFR1, reducing iron import and cell viability. In 3D culture or tumors this pressure is ostensibly relieved through upregulation of TFR1-independent iron internalization, allowing for pro-tumorigenic signals generated by AP2 loss to manifest.

## Discussion

We report that inactivating genes encoding the two unique subunits of AP2 universally reduced the growth in 2D culture of a panel of human PDAC cell lines, consistent with the common essential notation of these genes in DepMap[36,37]. We find that this reduction was attributed to disruption of iron transport. Namely, loss of AP2 prevented endocytosis of its cargo TFR1, and in turn iron import, which is an essential metal[57]. However, in 3D culture and tumors loss of AP2 either had no effect or even increased transformed or tumorigenic growth. In both these settings intracellular iron was not reduced, as evidenced by TFRC, FTH1, IREB2 and/or ACO1 levels, concomitant with a reprogramming of the iron transport genes *TFRC, SLC11A1, MCOLN1, SLC40A1 SLC22A17, STEAP3*, and *SCARA5* in favor of TF-independent iron internalization. We thus suggest that this allowed other effects of AP2 loss to manifest, namely enhanced tumor growth in a subset of the cell lines vis-à-vis retention of integrins on the plasma membrane leading to FAK activation and a corresponding proliferative transcriptional response (Fig. 6).

We note a number of points and caveats related to this model. First, it was previously established that TF endocytosis is not necessary for cell viability in all tissues in vivo. Specifically, while the *Tfrc^{-/-}* genotype is embryonic lethal, development is normal in both *Tfrc^{-/-}*

chimeric mice and upon tissue-specific ablation of this gene[58–62], consistent with TF-independent iron internalization sustaining cell proliferation in vivo. Iron can be absorbed independent of TF by at least ten pathways[38,43–50,63]. In agreement, we see evidence that iron import is restored in 3D culture and tumors, which occurs in lockstep with transcriptional reprograming that favors TF-independent iron internalization. These findings have implications for therapeutics targeting the transferrin receptor in cancer as a means to reduce cellular iron[19].

Second, it is unclear what aspect of the transition from 2D to 3D culture or tumors underlies the transcriptional changes in iron transport genes. However, upon AP2 depletion, cells grown in 3D exhibit iron import and enhanced growth as observed in tumors, suggesting the effect can be attributed to the less complex 3D culture environment. For example, mechanical properties, ECM-cell/cell-cell contact architecture, and soluble factor/oxygen gradients all differ between 2D and 3D culture[64–69].

Third, integrins were not only enriched at the plasma membrane upon AP2 depletion in tumors derived from BXPC-3 cells, but the total level changed as well, pointing to a more general disruption in trafficking beyond just endocytosis[16]. In the case of ITGA2 and ITGB6, the total levels increased. Perhaps then AP2 loss, which we document reduces endocytosis of ITGA2 and ITGB6 leading to accumulation at the plasma membrane, prevents the normal trafficking to the lysosome for degradation. In the case of ITGB1, the total levels decreased. Only a portion of ITGB1 endocytosis was AP2-dependent. Perhaps then the loss of AP2 shifts the proportion of ITGB1 to an AP2-independent pathway targeting the protein to the lysosome. Nevertheless, while the loss of AP2 may have more general effects on trafficking, the fact remains that integrins accumulated on the plasma membrane, which is known to lead to their activation.

Fourth, AP2 loss gradually enhanced tumor growth in the panel of cell lines, from no effect (e.g. CFPAC-1) to a level on par with loss of the tumor suppressor PTEN (e.g. BXPC-3). We took advantage of this gradient to interrogate the mechanism by which AP2 loss enhances tumor growth. Namely, we show that loss of AP2 altered the plasma membrane proteome in tumors regardless of the cell line, consistent with the known role of AP2 in endocytosis[70]. However, the transcriptome was only changed in cell lines in which tumor growth was enhanced, pointing towards a specific cargo being retained on the plasma membrane of these cells. Indeed, AP2 loss led to an accumulation of ITGB1 on the plasma membrane of tumors derived from BXPC-3 but not CPFAC-1 cells. Further, ITGB1 endocytosis was both AP2- and clathrin-dependent in BXPC-3 but not CFPAC-1 cells. Again however, we suggest that it is not that AP2-mediated endocytosis is absent in CFPAC-1 cells – loss of AP2 in these cells prevented

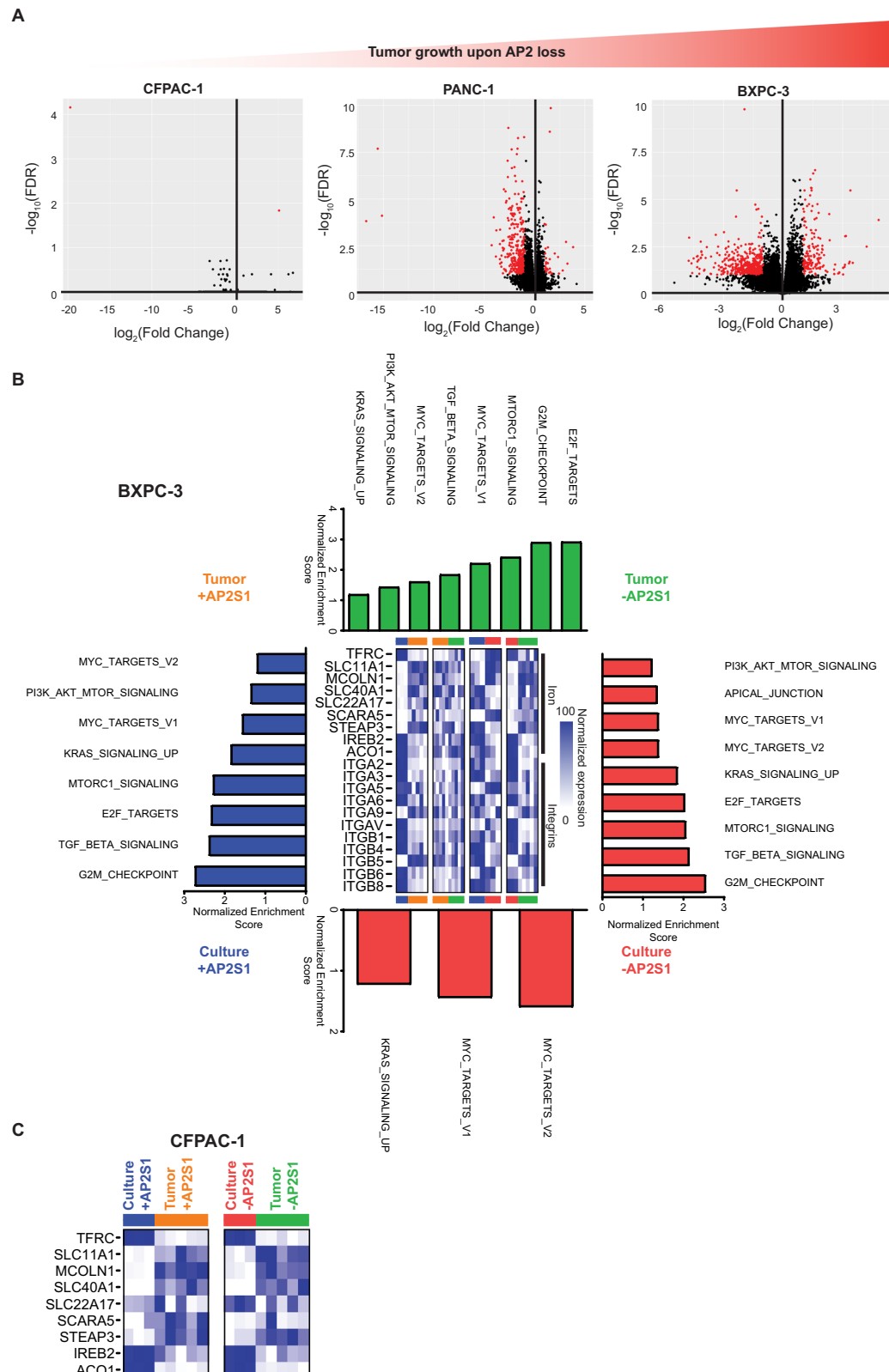

**Fig. 5 | Growth environment alters transcription to enhance oncogenic growth with AP2 loss of function. A** Volcano plot of log$_2$(fold change) versus -log$_{10}$(FDR) representing differential expression of transcripts comparing tumor samples from each cell type with and without *AP2S1* (red represents FDR < 0.05, log$_2$ FC > 1). Graphs are ordered by the enhanced tumor growth upon AP2 loss for each cell line. **B** Transcriptional changes in BXPC-3 cells as growth environment and *AP2S1* status are changed as determined by RNA sequencing. Outer bar graphs represent the absolute normalized enrichment scores (NES) of Hallmark gene sets with an

FDR < 0.25 and related to oncogenic proliferation. Central heat maps represent mean normalized expression of manually curated genes of interest from indicated pathways. (*n* = 5 tumors per condition and *n* = 3 technical cell culture replicates per condition). **C** Heat maps representing normalized mean CFPAC-1 expression determined by RNA sequencing of manually curated iron transport genes comparing cell culture to tumor samples (*n* = 5 tumors per condition and *n* = 3 technical cell culture replicates per condition). Source data are provided as a Source Data file.

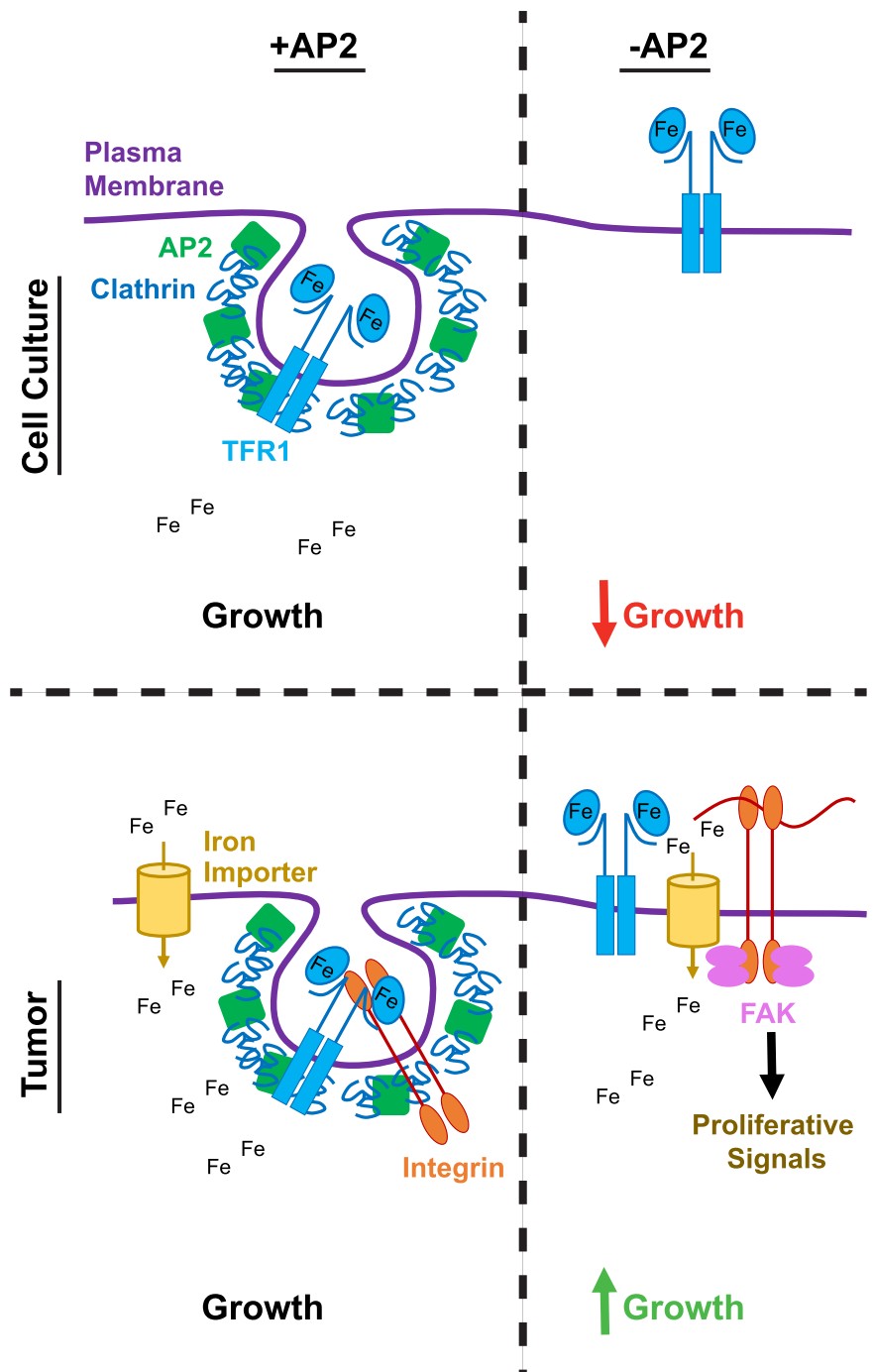

**Fig. 6 | A model of tumor microenvironment dependent effects of AP2.** In 2D culture cells are dependent upon AP2 for clathrin-mediated endocytosis of TFR1, TF, and bound iron (top left). As iron is an essential mineral, a loss of AP2 inhibits cell growth (top right). However, in tumors iron levels are restored and genes linked to internalization of alternate iron sources are upregulated (bottom left). This allows other proteins on the plasma membrane affected by AP2 loss to exert an effect, for example, accumulation of integrins to promote a proliferative signal (bottom right).

endocytosis of TFR1 – but rather that AP2 appears to discriminate between cargoes. Perhaps this may explain the discrepancies in whether a cargo is AP2-dependent or not contingent upon the cell line[71].

To conclude, we found that AP2 loss has completely oppositive phenotypes depending on the local environment, inhibiting versus enhancing growth in vitro versus in vivo. Whether tissue culture uniquely sensitized cancer cells to the loss of other genes remains unknown, but if the PtdIns sgRNA library is any indication, there may yet be more common-essential genes that oppositely impact tumorigenesis.

## Methods

All mouse care and experiments were performed in accordance with a protocol approved by the Institutional Animal Care and Use Committee (IACUC) of Duke University (protocol no. A195-19-09).

### Experimental Model and Subject Details

**Cell lines**. All cell lines were obtained from the Duke Cell Culture Facility and verified by STR profiling and maintained at 37 °C and 5% $CO_2$. HEK293t, PANC-1, and MIAPACA-1 cells were cultured in DMEM (High Glucose). BXPC-3 cells were cultured in RPMI1640. CFPAC-1 cells

were cultured in Iscove's modified Dulbecco's medium. HPAF-II cells were cultured in MEM supplemented with nonessential amino acids (1x) and sodium pyruvate (1 mM). All media was supplemented with 10% Fetal Bovine Serum (FBS)

**Mouse models.** Fox Chase SCID Beige mice (CB17.Cg-PrkdcscidLystbg-J/Crl, Charles River) were used in all studies. All mouse care and experiments were performed in accordance with a protocol approved by the Institutional Animal Care and Use Committee (IACUC) of Duke University (protocol no. A195-19-09). In accordance with Duke Institutional Animal Care and Use Committee approval no tumors exceeded 1.5 cm$^3$. Experiments began with mice within 6–8 weeks of age and humanely euthanized before 6 months by $CO_2$ asphyxiation with secondary euthanasia performed by decapitation, after which tumors were removed during necropsy. Both male and female mice were used in this study.

### Method details
**Compounds and inhibitors.** Unless otherwise noted, compounds and inhibitors were used at the following concentrations. Doxycycline (Millipore Sigma, Cat # D9891)- 1 µg/ml; Pitstop2 (Millipore Sigma, Cat# SML1169) – 30 µM; FAKi (PF-573228,Selleckchem, Cat# S2013) - 10 µM; Cilengitide (Selleckchem, Cat# S6387). Deferoxamine (Millipore Sigma, Cat# D0160000) – 100 µM; Ammonium Ferric Citrate (Millipore Sigma, Cat# F5879) – 100 µg/ml.

**in vivo and in vitro CRISPR/Cas9 screen**
**Library design.** The sgRNA library was designed to target all phosphatidylinositol synthetases, kinases, phosphatase, lipases and regulatory genes with 5 sgRNA each based on a previous report[72], in addition 100 control (ctrl) *sgRNAs* totaling 660 sgRNAs. See Supplemental Data 1 for sgRNA library details.

**Library cloning.** Pooled oligos encoding sgRNA with 5′ and 3′ adapter sequences (CustomArray Inc) were diluted 1:10 in water and amplified using Phusion Hotstart Flex master mix (New England Biolabs Inc) with the ArrayF and ArrayR primers (IDT).

ArrayF- - TAACTTGAAAGTATTTCGATTTCTTGGCTTTATATATCT TGTGGAAAGGACGAAACACCG

ArrayR- - ACTTTTTCAAGTTGATAACGGACTAGCCTTATTTTAACT TGCTATTTCTAGCTCTAAAAC

sgRNA amplicons were cloned into lentiCRISPR v2-Blast, a gift from Mohan Babu (Addgene plasmid # 83480; http://n2t.net/addgene: 83480; RRID:Addgene_83480). 5 µg of the lentiCRISPR v2-Blast vector was digested with FastDigest Esp3I (Thermo Fisher Scientific) enzyme and electrophoresed on a 1% agarose gel. The ~11 kb was excised from the gel and purified using a QIAquick Gel Extraction Kit according to the manufacturer's instructions (Qiagen). The amplified library was assembled into the restriction-digested vector using Gibson Assembly Master Mix (New England Biolabs Inc) and electroporated into library competent E. coli (Lucigen Corp). Library plasmid DNA was purified using a HiSpeed Plasmid Midi Kit according to the manufacturer's instructions (Qiagen) and library coverage was validated by Illumina NextSeq 500 single-end sequencing.

**Lentivirus production.** HEK293T cells were cultured to 70–80% confluency and transfected with lentiviral packaging system at the ratio of 1:1:2 for psPAX2:pVSVg:lentiCRISPR v2 Blast using Fugene6 (Promega) transfection reagent. After 48 to 72 hours, media was harvested from cells, centrifuged at 400 RCF for 3 minutes and filtered using a 45 µm pore filter (VWR) to remove cell debris. Virus was either used immediately or snap frozen for future use.

**Cell infection.** For each replicate, $1.2 \times 10^7$ cells were infected at an MOI of 0.3 to ensure that most cells were infected by a single lentiviral

particle. Each infection was performed across 6 wells of a 6-well plate in 1 ml of the appropriate media containing virus at a predetermined concentration to reach the targeted MOI in the presence of 8 µg/ml polybrene (MilliporeSigma) and spun at $800 \times g$ for 1 h. After 6 h of culture at 37 °C, cells were washed in 1x P̲hosphate B̲uffered S̲aline (PBS; Sodium Chloride, 0.137 M; Potassium Chloride, 0.0027 M; Sodium Phosphate Dibasic, 0.01 M; Potassium Phosphate Monobasic, 0.0018 M), and the media was replaced. 18 h later, each infection was pooled into a 15 cm culture dish and selected with 5 µg/ml blasticidin (Thermo Fisher Scientific), except for PANC-1 cells, which were selected with 20 µg/ml blasticidin. Cells were continuously passaged and selected for 1 week after initial infection at which point $1 \times 10^6$ cells were pelleted and snap frozen as the initial sample. $1 \times 10^6$ cells were continuously passaged for 2 weeks for in vitro samples. $5 \times 10^6$ cells were transplanted into 1 male and 1 female Fox Chase SCID/Beige mouse (details below) (Charles River). In vitro samples were pelleted and snap frozen after 2 weeks of growth.

**Xenograft transplant.** For each transplant, $5 \times 10^6$ cells were resuspended in 200 µl sterile PBS and kept on ice until injection (less than 2 h). Mice were anesthetized in a bell jar with diffuse isoflurane, their injection site sterilized with iodine, and subcutaneously injected with 200 µl cell suspension in the right flank using a 30 G needle (BD). Mice were monitored hourly for 24 h, then afterward, three times a week for tumor growth and morbidity and moribundity. Mice were humanely euthanized by $CO_2$ asphyxiation followed by decapitation upon a tumor size of ~1 cm$^3$ or upon signs of moribundity, and tumors removed and snap frozen during necropsy.

**gDNA isolation, sgRNA amplification, and sequencing.** Cell samples were lysed in 1 ml DNAzol by vigorous pipetting according to the manufacture's protocol (Molecular Research Center inc.). DNA was precipitated in 0.5 ml 200 proof ethanol (VWR). 30 to 50 mg of tumor samples were lysed in 5 ml DNAzol by a motorized tissue homogenizer (Kinematica AG PT1200), centrifuged at 3,200 g for 15 minutes at 4 °C, and transferred to a new tube. gDNA was then precipitated by 2.5 ml 200 proof ethanol. Precipitated DNA was pelleted by centrifugation at 20,000 g, washed twice in 75% ethanol. gDNA was solubilized in 0.5 ml or 1.0 ml of water for cell and tumor samples, respectively. To reduce PCR bias, amplification of sgRNA sequences from each sample gDNA was performed in triplicate and pooled at each step. First round of PCR amplification was performed using 50 to 100 µg gDNA, Hotstart Taq (Takara), and 1 µM CRISPR F1 and CRISPR R1 primers (IDT).

CRISPR F1 - AATGGACTATCATATGCTTACCGTAACTTGAAAGTAT TTCG

CRISPR R1 - TCTACTATTCTTTCCCCTGCACTGTtgtgggcgatgtgcg ctctg

Thermal cycles consisted of 95 °C for 5 minutes, followed by 18 cycles of 95 °C for 30 seconds, 60 °C for 30 seconds, 72 °C for 30 seconds, and a final cycle of 72 °C for 5 minutes. Amplicons from the same samples were pooled and a second round of PCR amplification was performed to add sample barcode and Illumina adapter sequences using 20 µl of PCR1 and 0.5 µM CRISPR R2 and barcoded forward primers. See Supplemental Data 2 for barcode information.

CRISPR R2 - CAAGCAGAAGACGGCATACGAGATGTGACTGGAGT TCAGACGTGTGCTCTTCCGATCTtctactattctttcccctgcactgt

Thermal cycles consisted of 95 °C for 5 minutes, followed by 20 cycles of 95 °C for 15 seconds, 58 °C for 30 seconds, 72 °C for 30 seconds, and a final cycle of 72 °C for 5 minutes. Amplicons from the second round of PCR were pooled at equal concentrations, were purified using Ampure XP Beads according to the manufacture's protocol (Beckman Colter). Purified amplicons were subjected to Illumina NextSeq 500 75 bp single end sequencing.

**Individual *sgRNA* cloning.** Individual *sgRNA*s were cloned into the TLCV2 vector, a gift from Adam Karpf (Addgene plasmid # 87360; http://n2t.net/addgene:87360; RRID:Addgene_87360)[29] identical to the protocol used to clone the sgRNA library into the lentiCRISPR v2–Blast vector. *sgRNA* oligo sequences were:

AP2S1 sgRNA1 - GGAAAGGACGAAACACCGGATCGAGGAGGTGC ATGCCGGTTTTAGAGCTAGAAATAGCAAGTTAAAATAAGGC

AP2S1 sgRNA2 - GGAAAGGACGAAACACCGCAAACACACCAA CTTTGTGGGTTTTAGAGCTAGAAATAGCAAGTTAAAATAAGGC

AP2M1 sgRNA - GGAAAGGACGAAACACCGACTGCTGGCTCAGAA GATCGGTTTTAGAGCTAGAAATAGCAAGTTAAAATAAGGC

PTEN sgRNA - GGAAAGGACGAAACACCGGAACTTGTCTTCCC GTCGTGGTTTTAGAGCTAGAAATAGCAAGTTAAAATAAGGC

**TRIPZ *shRNA* clones.** TRIPZ-*shRNA* clones were purchased from Horizon Discovery. AP2M1 was targeted using two different clones based on maximal knockdown efficiency. In BXPC-3 cells clone V2THS_172835 (mature antisense sequence: TAAAGTTGGACTTGAT-GAC)was used while in CFPAC-1 clone V3THS_304543 (mature antisense sequence: TGGACTTGATGACCACCTT) was used.

**Cloning TRIPZ-GFP.** TRIPZ-GFP was produced by digestion of FUGW, a gift from David Baltimore (Addgene plasmid # 14883; http://n2t.net/addgene:14883; RRID:Addgene_14883)[73] and TRIPZ-RFP (Horizon Discovery) with AgeI and EcoRI restriction enzymes (New England Biolabs Inc) to excise the DNA encoding RFP from TRIPZ and the GFP from FUGW. GFP and TRIPZ sequences were then gel purified with Gel Extraction kit (Qiagen) as above, ligated, introduced into bacteria by transformation, and purified using standard protocols. The identity of the plasmids was confirmed by Sanger sequencing.

**Plasmid availability.** Plasmids generated in this study has been deposited with Addgene(TLCV2 - sg*AP2S1* #1, Addgene #202756; TLCV2 - sg*AP2S1* #2, Addgene #202757; TLCV2 - sg*AP2M1*, Addgene #202758; TLCV2 – sgPTEN, Addgene #202759).

**Lentivirus production and cell infection.** Lentivirus production and cell infection were performed similar to generating lentiviral sgRNA libraries with the following exceptions. HEK293t cells were transfected with variable transfer plasmids encoding the shRNA (pTRIPZ; Horizon Discovery) or sgRNAs (TLCV2; (Addgene plasmid # 87360; http://n2t.net/addgene:87360; RRID:Addgene_87360)[29]) of interest. Cell infections were performed by infection with $2 \times 10^6$ cells/well of a 6-well plate with variable viral MOI. 24 h after infection, cells were selected with 2 μg/ml puromycin for 48 h, except in the case of PANC-1 cells, which were selected with 4 μg/ml.

**Doxycycline-inducible CRISPR/Cas9 knockout.** BXPC-3, PANC-1, and CFPAC-1 cells were transduced with TLCV2 lentivirus encoding doxycycline-inducible Cas9 and *sgRNA*s targeting *AP2M1*, *AP2S1*, or *PTEN*. After 48 h of puromycin selection, cells were induced with 1 μg/ml of doxycycline for 7 days unless stated otherwise. Effects on protein expression were validated by immunoblot. For CFPAC-1 cells this protocol was slightly modified to increase knockout efficiency. During doxycycline induction, cells were grown I the presence of AFC. This relieved negative selection against cells where knockout had occurred by supplying cells with iron necessary for proliferation.

**Doxycycline-inducible *shRNA* knockdown.** BXPC-3 cells were transduced with pTRIPZ lentivirus encoding doxycycline-inducible *shRNA* targeting *AP2M1* or a non-targeting control. After 48 hours of puromycin selection, cells were induced with 1 μg/ml of doxycycline for 48 h before the beginning of an experiment unless stated otherwise. Effects on protein expression were validated by immunoblot.

**in vitro growth assays.** $1 \times 10^4$ or $5 \times 10^4$ cells/well were plated in 12- or 6-well plates respectively. Cells were grown for 4 days, media was changed, then 3 days later wells were washed with 1x PBS and cells were fixed in 10% neutral buffered formalin (VWR) for 5 minutes at room temperature. Each well was then stained with 0.5% crystal violet (MilliporeSigma) for 10 minutes, washed 3 times with distilled water and allowed to dry overnight. Plates were imaged on a flatbed scanner at 600dpi resolution (EPSON).

**Xenograft growth assays.** Cells were transplanted into the right flank of female SCID/beige mice as xenografts as detailed in the CRISPR screen method. Tumor volumes were measured every 3 times a week until endpoint (volume -1 cm³ or at moribundity). Tumor volumes were estimated using by calculating (length*width²)/2.

**Endocytosis assays.** Cells transduced with pTRIPZ lentivirus encoding RFP and *shCTRL* or *shAP2M1* were plated and transduced with 1 μg/ml doxycycline. 72 hours later cells were serum starved for 1 hour in the presence of doxycycline and then treated with Pitstop2 or an equivalent volume of DMSO for an additional hour. Cells were placed on ice for 5 min and treated with CF488-Transferrin (Biotium, 10 μg/ml) or Alexa488 conjugated Integrin β1 antibody (Santa Cruz, clone 12G10, 1:250) and incubated at 37 °C for 10 or 30 minutes respectively. Cells were then washed in ice cold 0.2 M Acetic Acid + 0.5 M NaCl for 10 min on a rocker to remove extracellular fluorescence and fixed in 4% paraformaldehyde (Thermo Fischer Scientific) in PBS. Cells were subsequently permeabilized in 0.1% TritonX-100 in PBS and stained with DAPI.

**Tide analysis.** gDNA was isolated from tumors and cell samples using QiaAmp DNAmini kit as above. Amplicons were then generated using primers -400 base pairs 5' and 3' of the *AP2S1 sgRNA* target site (TideAP2S1-F and TideAP2S1-R) using MyTaq Red 2x master mix (Bioline).

TideAP2S1-F - TGGCTTGGGTCTGGGGAG

TideAP2S1-R - CACCTGGCCTATTCTCTC TAC T

Thermocycle consisted of 95 °C for 2 minutes, 34 cycles of (95 °C for 30 seconds, 60 °C for 30 seconds, and 72 °C for 1 minute, followed by 72 °C for 5 minutes. A single -850 bp amplicon was generated for each sample. Amplicons were purified using PCR purification kit according to the manufacturer's instructions (Qiagen) and sequenced by the Sanger method. Chromatograms from each sequence were subjected to *T*racking of *I*ndels by *DE*composition (TIDE)[31] analysis to determine insertion and deletion efficiency.

**Spheroid assays**

**Production of Methylcellulose stock media.** Methylcellulose stock media is necessary to efficiently generate spheroids. Stocks were made by autoclaving 1.2 g of 4000 cp methylcellulose with a magnetic stirbar in 100 ml glass bottle. 50 ml base media heated to 60 °C was added to cooled methylcellulose and stirred for 30 minutes. 40 ml of base media, 10 ml FBS were added and further supplemented with penicillin (100 U/ml), and streptomycin (100 mg/ml). Stock media was stirred for 16 h at 4 °C. Media was then moved to two 50 ml conical tubes and centrifuged at 800 g for 60 minutes. 10 ml aliquots were frozen at −20 °C.

**Hanging droplet spheroid growth.** $1 \times 10^4$ BXPC-3 cells transduced with pTRIPZ doxycycline-inducible RFP and non-targeting or *AP2M1* targeting *shRNA* were resuspended in 125 μl of methylcellulose stock media supplemented with 1 μg/ml doxycycline and brought to 1 ml total volume with RPMI1640 supplemented with 10% FBS. Each spheroid was formed by 20 μl ( - 200 cells) in 1 micro-well of a 60 micro-well plate. Spheroids were cultivated by inverting the plate in a humidity chamber at 37 °C and 5% CO₂. In most wells an individual

spheroid formed at the media-air interface after 48 h. Only wells exhibiting spheroids were analyzed. Every 3 days, 10 µl of RPMI1640 / 10% FBS was added to each well. Spheroids were imaged every 24 h after initial plating for 11 days. Spheroids were imaged using a Leica binocular inverted microscope equipped with a 10x objective and camera.

**Growth competition.** 12-well plates were coated in 0.5 ml 1.5% agarose in PBS gel per well. Cells were transduced with pTRIPZ lentivirus encoding doxycycline-inducible RFP-*shCTRL*, GFP-*shCTRL*, or RFP-*shAP2M1*. $5 \times 10^5$ total cells were resuspended as 50:50 mixtures (RFP-*shCTRL*: GFP-*shCTRL*, RFP-*shAP2M1*, or GFP-*shCTRL*) in 8 ml total volume (1 ml methylcellulose stock media, 1 µg/ml doxycycline, base media / 10% FBS to 8 ml). 1 ml of each mixture was plated in 1 well of a gel coated (3-D) and uncoated (2-D) well of a 12-well plate per replicate. After 5 days, 1 ml of base media supplemented with 10% FBS and 1 µg/ml doxycycline was added to each well. 7 days after plating, samples were imaged using Evos M5000 epifluorescence microscope with RFP and GFP fluorescence filter sets and light sources. To determine RFP/GFP cell ratios, cells and spheroids were washed 3 times with PBS, dissociated with 0.05% trypsin-EDTA, and counted using a Countess IIFL cell counter equipped with RFP and GFP fluorescence filter sets and light sources.

**Spheroid immunofluorescence staining.** 10 cm dishes were coated with 5 ml of 1.5% agarose in PBS gel. $2 \times 10^6$ BXPC-3 cells transduced with pTRIPZ lentivirus encoding RFP and *shAP2M1* were plated in coated dishes with 1.25 ml methylcellulose stock media, 1 µg/ml doxycycline, and RPMI1640 / 10% FBS to a total volume of 10 ml. After 4 days, spheroids were pelleted by centrifugation at 400 RCF for 3 minutes. Between each of the following steps, spheroids were pelleted in the same manner. Spheroids were then washed in PBS, fixed in 4% paraformaldehyde (Thermo Fisher) in PBS for 30 minutes, washed again in PBS, and blocked with 5% bovine serum albumin (BSA) in PBS for 16 h at 4 °C. Spheroids were subsequently incubated on a rotator with an anti-integrin β6 antibody (Cell Signaling) which recognizes extracellular epitopes of the protein at a 1:200 dilution in 5% BSA/PBS for 24 h at 4 °C. After incubation with primary antibody, spheroids were washed 3 times in PBS and incubated on a rotator with anti-rabbit IgG conjugated to Alexa-488 dye (Thermo Fisher) at a 1:1000 dilution in 5% BSA/PBS for 24 h at 4 °C. After secondary antibody incubation, spheroids were washed 3 times in PBS, resuspended in fluoromountG (Thermo Fisher) mounting medium and affixed between a slide and cover glass. Spheroids were then imaged with an Evos M5000 (Thermo Fisher) epifluorescence microscope equipped with RFP and GFP fluorescence filter sets and light sources.

**2D and 3D RT-qPCR.** 6 cm dishes were coated with 3 ml of 1.5% agarose in PBS. For each comparison $3.125 \times 10^5$ cells were plated on one agarose and one tissue culture treated dish in 5 ml medial containing 0.625 ml methylcellulose stock media. Cells were incubated for 4 days before purifying RNA from samples using RNeasy plus mini (Qiagen) according to the manufacturer's protocol. RT-qPCR was performed using GoTaq 1-step RT-qPCR (Promega) according to manufacturer's directions and assayed in a 384-well 10 µl volume format using a Biorad CFX384. RT-qPCR primers were generated using PrimerBank[74] (https://pga.mgh.harvard.edu/primerbank/) and can be found in Supplemental Data 6.

**Iron supplement rescue.** $1 \times 10^4$ cells transduced with pTRIPZ lentivirus encoding RFP and *shAP2M1* were plated in each well of two 24-well plates in 0.5 ml media. Each well of one plate was treated with 1 µg/ml doxycycline. Four wells of each plate were treated with 0, 1, 10, 100, 200, or 500 µg/ml ammonium ferric citrate (MilliporeSigma) for technical replicates. After 4 days, media in each well was changed following

the same conditions. At 7 days cells were fixed, stained, and analyzed as previously described in the in vitro growth assay methods.

**Mitochondrial iron analysis by Mito-ferro green staining.** BXPC-3 cells transduced with pTRIPZ lentivirus encoding RFP and *shCTRL* or *shAP2M1* were plated and transduced with 1 µg/ml doxycycline. 72 h later cells were dissociated from the plate by trypsin/EDTA solution and each plated in a 12 well plate with 1 µg/ml doxycycline and allowed to adhere for 8 h. Each well was then washed with PBS and treated with 100 µM deferoxamine (Millipore Sigma) in serum free media for 1 h. Wells were again washed with PBS and treated with 100 µM deferoxamine, 100 µg/ml ammonium ferric citrate or 100 µg/ml holo-transferrin (Thermo Fisher Scientific) for 16 h in serum free media. Cells sterile Hank's Balanced Salt Solution (HBSS) (Thermo Fisher Scientific) at 37 °C and then incubated with 5 µM Mito-ferro green (Fisher Scientific) in HBSS for 1 h. Each plate was washed again 3 times in HBSS and immediately imaged on an EVOS M5000 epi-fluorescent microscope equipped with RFP and GFP filter sets and light source.

**Deferoxamine dose response.** BXPC-3 cells transduced with TRIPZ-RFP *shCTRL* or *shAP2M1* were treated with 1 µg/ml of doxycycline to induce expression of *shRNA* and RFP and grown for 48 h. 5000 doxycycline -treated cells/well were plated in a 96-well plate in triplicate for 8 treatments (total = 24 wells/ cell type) with 50 µl of media with 1 µg/ml doxycycline. 50 µl of deferoxamine diluted in media with 1 ug/ml doxycycline was then added in triplicate to both cell lines at the following concentrations: 1,562.5, 312.5, 62.5, 12.5, 2.5, 0.5, 0.1, and 0 µM. 3 columns of blank wells with 100 µl media and 1 µg/ml doxycycline were used as a background control. Cells were grown for 72 h, then measured with CellTiter-Glo Luminescent Cell Viability Assay (Promega) using a plate reader with PMT dwell time set to 0.5 seconds.

**RNA sequencing**

**mRNA isolation and library preparation.** $1 \times 10^6$ cells and 30 mg tumor samples were snap frozen in RNA Later solution (Thermo Fischer Scientific). At time of mRNA isolation, all equipment and work areas were cleaned with 70% ethanol of RNAse Away (Thermo Fisher Scientific). mRNA was isolated from samples using RNeasy Plus Mini kits following the manufacturer's instructions (Qiagen). Extracted total RNA quality and concentration were assessed on Fragment Analyzer (Agilent Technologies) and Qubit 2.0 (Thermo Fisher Scientific), respectively. All samples had an RQN > 7. RNA-seq libraries were prepared using the commercially available KAPA Stranded mRNA-Seq Kit (Roche) using 500 ng of total RNA for each sample. Illumina sequencing adapters were ligated to the output cDNA fragments and amplified to produce the final RNA-seq library with a dual-indexing approach for pooled library multiplex sequencing.

**Sequencing.** Before pooling and sequencing, each sub-library was assessed by a Fragment Analyzer (Agilent) and Qubit (Thermo Fisher Scientific) to determine fragment length distribution and Molarity. All libraries were then pooled in equimolar ratio and sequenced by Illumina NovaSeq 6000 sequencer.

**Cell membrane proteomics**

**Cell membrane protein enrichment.** 30–50 mg tumor samples were snap frozen. Frozen samples were thawed on ice and processed using Minute Plasma Membrane Protein Isolation and Cell Fractionation Kit following the manufacturer's instructions (Invent Biotechnologies). Samples were supplemented with 8 M urea and protein concentrations were determined via Bradford assay, ranging from 0.4877–1.064 µg/µl.

**Sample preparation for mass spectrometry.** Samples were normalized to 15 µg protein using 8 M urea and spiked with undigested bovine casein at a total of either 100 or 200 fmol/µg as an internal quality

control standard. Samples were then supplemented with 7.9 µl of 20% SDS, reduced with 10 mM dithiothreitol for 45 minutes at 32 °C, alkylated with 20 mM iodoacetamide for 30 minutes at room temperature, then supplemented with a final concentration of 1.2% phosphoric acid and 477 µl of S-Trap (Protifi) binding buffer (90% MeOH/100 mM _Tri_ethyl_a_mine _B_uffer (TEAB)). Proteins were trapped on the S-Trap micro cartridge (Protifi), digested using 20 ng/µl sequencing grade trypsin (Promega) for 1 h at 47 °C, and eluted using 50 mM TEAB, followed by 0.2% formic acid, and lastly using 50% acetonitrile/0.2% formic acid. All samples were then lyophilized to dryness. Samples were resolubilized using 30 µ; of 1% trifluoroacetic acid /2% acetonitrile with 12.5 fmol/µl yeast alcohol dehydrogenase.

**Mass spectrometry.** Quantitative liquid chromatography with tandem mass spectrometry (LC/MS/MS) was performed on 2 µl using an MClass UPLC system (Waters Corp) coupled to a Orbitrap Fusion Lumos high resolution accurate mass tandem mass spectrometer (Thermo Fisher Scientific) equipped with a FAIMSPro device via a nanoelectrospray ionization source. Briefly, the sample was first trapped on a Symmetry C18 20 mm × 180 µm trapping column (5 µl/minute at 99.9/0.1 v/v water/acetonitrile), after which the analytical separation was performed using a 1.8 µm Acquity HSS T3 C18 75 µm × 250 mm column (Waters Corp) with a 90-minute linear gradient of 5 to 30% acetonitrile with 0.1% formic acid at a flow rate of 400 nl/minute with a column temperature of 55 °C. Data collection on the Fusion Lumos mass spectrometer was performed for three difference compensation voltages (−40v, −60v, −80v). Within each CV, a data-dependent acquisition (DDA) mode of acquisition with a r = 120,000 (@ m/z 200) full MS scan from m/z 375–1500 with a target AGC value of 4e5 ions was performed. MS/MS scans were acquired in the ion trap in Rapid mode with a target AGC value of 1e4 and max fill time of 35 milliseconds. The total cycle time for each CV was 0.66 seconds, with total cycle times of 2 seconds between like full MS scans. A 20-second dynamic exclusion was employed to increase depth of coverage. The total analysis cycle time for each injection was ~2 h.

**Immunoblotting and Coomassie staining.** Cell samples were washed 3 times in PBS to remove serum proteins and lysed by scraping cells in _R_adio_immunop_recipitation _A_ssay (RIPA, Tris Chloride, 10 mM; EDTA 1 mM; EGTA, 0.5 mM; Triton X-100, 1%; Sodium Deoxycholate, 0.1%; SDS, 0.1%; Sodium Chloride, 140 mM) buffer supplemented with Complete Protease Inhibitor Cocktail (Roche) and 50 mM sodium fluoride. Tumor whole cell lysates were generated by lysing a 30 to 50 mg tumor sample on ice in 500 µl the same buffer using a mechanical tissue homogenizer. All lysates were rotated at 4 °C for 30 minutes and centrifuged at 4 °C for 15 minutes at 21,000 g to clear lysate. Tumor cell membrane proteins were enriched by Minute Plasma Membrane Protein Isolation and Cell Fractionation Kit (Invent Biotechnologies). Lysate protein concentrations were assayed by Bicinchoninic Acid solution assay (Millipore Sigma) and read on a Glomax Multidetection system (Promega). Lysates were resolved by 18% SDS-PAGE, transferred to a PVDF membrane (Bio-Rad, 1704273), blocked in 5% milk or for phosphorylation specific antibodies 5% BSA, and immunoblotted by the following antibodies in 5% milk or 5% BSA (Sigma, A7906-500G): and ß-Tubulin (Sigma, T5201; diluted 1:5,000), PTEN (Cell Signaling, 9552; 1:1000), RPS16 (Abcam, ab26159; 1:2500), AP2S1(Abcam, ab128950; 1:250), _AP2M1_(Abcam, ab75995; 1:250), GapDH(Santa Cruz, sc-365062; 1:5000), ITGB1 (Cell Signaling, 9699; 1:1000), FAK (Cell Signaling, 3285; 1:1000), pFAK (Cell Signaling, 8556; 1:1000), FTH1 (Cell Signaling, 4393; 1:500), TFRC (Cell Signaling, 13113; 1:500), IREB2 (Cell Signaling, 37135 s; 1:500), ITGA2 (Thermo Fisher Scientific, MA5-32306; 1:500), ITGA3 (Thermo Fisher Scientific, MA5-28565; 1:250), or ITGB6 (Cell Signaling, 95153; 1:1000). Primary antibody incubation was performed at 4 °C overnight followed by the secondary antibody incubation for 1 h at room temperature.

**Manuscript editing by AI.** ChatGPT 3.5 was used to make the text more concise.

## Quantification and statistical analysis

**CRISPR screen sequencing analysis.** Sequencing reads were demultiplexed using fastx_barcode_splitter (v0.0.13) to identify inline barcodes and aligned to the reference library using bowtie (v1.0.0), allowing for 0 mismatches. Sample sgRNA read counts were first normalized to total reads/sample. sgRNA fold-change values were then calculated by comparing in vitro and in vivo sample sgRNA counts to their paired initial sample sgRNA count. Gene target enrichment scores were then calculated by averaging the 5 sgRNA fold-change values within a sample. Average sample enrichment fold-changes distributions were normalized by first zeroing each sample on the average enrichment of non-targeting control s_gRNA_s and then calculating a z-score. To compare in vitro enrichment to in vivo enrichment, replicate sample z-scores were averaged and compared by T-test. Resultant _p_-values were adjusted by calculating false discovery rates.

**In vitro growth assay analysis.** Plate images were quantified using ImageJ[75] (1.53t) image analysis software. Regions or interest encompassed the entire well, thresholds were set based on the green channel values from color images and set consistently across all wells analyzed to produce a binary mask. The percent of well area covered in cells was then calculated as masked area/total well area. Relative growth areas were calculated by comparing to the average control value. Prism (9.5.1) software was used to perform t-tests comparing growth to control samples.

**Xenograft growth comparison analysis.** Individual xenograft volume data sets for each day of measurement were compared by 2-way ANOVA with a muti-comparison post-test in Prism (9.5.1) software.

**Spheroid immunofluorescence staining analysis.** Image analysis and quantification was performed in ImageJ (1.53t) software. Alexa-488 fluorescent intensity was determined for each image by manual segmentation of spheroid areas using a phase contrast image followed by integrated intensity measurement of the green channel within the segmented area. Mean pixels intensity values were calculated to account for differences in spheroid size. Mean intensities were compared by t-test using Prism (9.5.1) software.

**Spheroid growth analysis.** Spheroid area was assessed by manually outlining spheroid images from phase-contrast micrographs using ImageJ (1.53t) software. Assuming spheroids were spherical, relative volumes were estimated by comparing measured area at 11 days to 2days. Differences in average relative volumes were assessed by t-test using Prism (9.5.1) software.

**Spheroid growth competition analysis.** Differences in average relative GFP/RFP ratios were assessed by t-test using Prism (9.5.1) software.

**Mito-ferro green staining analysis.** Image analysis and quantification was performed in ImageJ (1.53t) software. Mito-ferro green fluorescence/RFP[+] cells were quantified from micrographs by first segmenting by thresholding the image on the RFP channel to determine the RFP[+] cell area. Integrated intensity values of the GFP channel were then measured for each RFP[+] cell area. Sample means were compared by 1-way ANOVA with multi-comparison post-test in Prism (9.5.1) software.

**Dose response curve analysis.** Background control well values were subtracted from test values, then replicate experiment values were normalized to maximum values and combined. Non-linear regression analysis was used to fit a curve to each data set using GraphPad Prism

(9.5.1). The Area Under the Curve (AUC) statistics were then calculated using Prism. Statistical comparison of AUCs were made by t-test.

**RNA sequencing data analysis.** RNA-seq data was processed using the TrimGalore (0.6.5) toolkit which employs Cutadapt to trim low-quality bases and Illumina sequencing adapters from the 3′ end of the reads. Only reads that were 20 nucleotides or longer after trimming were kept for further analysis. Reads were mapped to the GRCh38v93 version of the human genome and transcriptome using the STAR (2.7) RNA-seq alignment tool. Reads were kept for subsequent analysis if they mapped to a single genomic location. Gene counts were compiled using the HTSeq (2.0.3) tool. Differential gene expression was assessed using R version 4.2.1 coupled with DESeq2 (1.40.1), ggplot2 (3.4.2), dplyr (1.1.2) and pheatmap (1.1.2) libraries. Gene Set Enrichment Analysis[76] was performed on the normalized counts output by DEseq2 using the GSEA (4.3.2) software package. The h.all.v2023.Hs.symbols.gmt gene sets from the Molecular Signatures Database were used for the analysis. Gene symbols were collapsed, permutations were performed on gene_set rather than phenotype and the chip platform selected was the Human_Ensembl_Gene_ID_MSigDB. v2023.1.Hs.chip.

**RT-qPCR data analysis.** Cq values were exported from BioRad CFX384. ΔCq values were calculated by subtracting the matched Cq value of GAPDH PCR1 from the Cq of the gene of interest. Mean ΔCq values were used to compare 2D to 3D growth conditions and t-tests statistical analysis was performed In Prism (9.5.1). Expression differences are reported as $2^{(-\Delta\Delta Cq)}$.

**Proteomics data analysis.** Following UPLC-MS/MS analyzes, data were imported into Proteome Discoverer 2.5 (Thermo Fisher Scientific Inc). In addition to quantitative signal extraction, the MS/MS data was searched against the SwissProt *H. sapiens* database (downloaded in Nov 2019) and a common contaminant/spiked protein database (bovine albumin, bovine casein, yeast ADH, etc.), and an equal number of reversed-sequence "decoys" for false discovery rate determination. Sequest with Infernys enabled (v 2.5, Thermo PD) was utilized to produce fragment ion spectra and to perform the database searches. Database search parameters included fixed modification on Cys (carbamidomethyl) and variable modification on Met (oxidation). Search tolerances were 2 ppm precursor and 0.8 Da product ion with full trypsin enzyme rules. Peptide Validator and Protein FDR Validator nodes in Proteome Discoverer were used to annotate the data at a maximum 1% protein false discovery rate based on q-value calculations. Peptide homology was addressed using razor rules in which a peptide matched to multiple different proteins was exclusively assigned to the protein has more identified peptides. Protein homology was addressed by grouping proteins that had the same set of peptides to account for their identification. A master protein within a group was assigned based on% coverage. Prior to imputation, a filter was applied such that a peptide was removed if it was not measured in at least 2 unique samples (50% of a single group). After that filter, any missing values were imputed using the following rules: 1) if only one single signal was missing within the group of three, an average of the other two values was used or 2) if two out of three signals were missing within the group of three, a randomized intensity within the bottom 2% of the detectable signals was used. To summarize to the protein level, all peptides belonging to the same protein were summed into a single intensity. These protein levels were then subjected to a normalization in which the top and bottom 10% of the signals were excluded and the average of the remaining values was used to normalize across all samples. In order to assess processing and technical reproducibility, we calculated% coefficient of variation (%CV) across replicate Serum Pooled Quality Control sample injections as well as within each biological group. The average %CV of all proteins across the replicate

SPQC injections was 10.0%. The average %CV of all proteins within each unique biological group was 23.1%. Fold-changes between sample groups were calculated based on the protein expression values and calculated two-tailed heteroscedastic t-test on log2-transformed data for this comparison. Gene Ontology enrichment analysis was performed using the g:Profiler online tool with proteins that were enriched with an absolute fold-change value greater than 1.5 and FDR < 0.05.

**DepMap data analysis.** To assess transcriptional variability in PDAC cell lines by Principle Component Analysis (PCA), OmicsExpressionProteinCodingGenesTPMLogp1.csv was downloaded on 4-5-2023 as part of the DepMap Public 22Q4 data set. Transcript expression PCA analysis was performed in R studio (2023.3.0 build 386) with R (4.2.1) using ggplot2 (3.4.2) and dplyr (1.1.2) libraries. Gene Effect and Co-dependency statistics were accessed from the online data portal on 3-15-23 for the purpose of figure production.

### Reporting summary
Further information on research design is available in the Nature Portfolio Reporting Summary linked to this article.

## Data availability
Source data are provided with this paper. The RNA-seq data generated in this study have been deposited in the GEO database under accession code GSE233173. The PtdIns targeted CRISPR screen data generated in this study have been deposited in the GEO database under accession code GSE289756. The proteomics data generated in this study have been deposited in the ProteomeXchange database under accession code PXD042404. OmicsExpressionProteinCodingGenesTPMLogp1.csv was downloaded on 4-5-2023 as part of the DepMap Public 22Q4 data set. The remaining data are available within the Article, Supplementary Information or Source Data file. Source data are provided with this paper.

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

## Acknowledgements

This work was supported by the National Cancer Institute (R01CA269272 to CMC, T32CA009111 and F32CA236183 to SPZ) and the Shared Resources of the Duke Cancer Institute (P30CA014236).

## Author contributions

Conceptualization, S.P.Z. and C.M.C; Methodology, S.P.Z. and C.M.C; Validation, S.P.Z. and C.M.C.; Forma Analysis, S.P.Z. and L.B.D.; Investigation, S.P.Z. and L.B.D; Data Curation, S.P.Z. and L.B.D.; Writing–Original Draft, S.P.Z; Writing–Review & Editing, S.P.Z., L.B.D., and C.M.C.; Visualization, S.P.Z.; Funding Acquisition, S.P.Z. and C.M.C.

## Competing interests

C.M.C. is co-Founder of Merlon Inc, a member of the External Advisory Board for the University of Colorado Cancer Center, has a cross appointment with Duke-NUS, is ex-officio of the executive team for the Cancer Biology Training Consortium, has previously consulted for the Guidepoint Network in an ad hoc fashion, and received licensing reimbursement from Humacyte Inc. None of these relationships provided salary or research support, nor did they play a role in the study design, data collection, data analysis, or decision to publish for this project. The remaining authors have no conflicts to disclose.
