## [Transparent Peer Review file · Nature Communications]

The essential clathrin adaptor protein complex-2 is tumor suppressive specifically in vivo

Corresponding Author: Professor Christopher Counter

Version 1:

Reviewer comments:

Reviewer #1

(Remarks to the Author)

Review comments:

The authors pointed out a long-term mystery in cell biology that the phenotypes of loss-of-function of genes are inconsistent in vitro versus in vivo. The authors utilized multiple genetic and bioinformatic approaches to investigate the question and focused on phosphatidylinositol metabolism. They identified that AP2 depletion demonstrates the opposite phenotypes in cell growth in vitro and in vivo in pancreatic ductal adenocarcinomas. They propose a working model that the AP2 loss reduces iron uptake to impair cell growth in vitro, however, the retention of integrins on the plasma membrane owing to AP2 loss activates proliferative signaling (e.g. PI3K/AKT) to promote tumor growth in vivo. They also provide evidence that the AP2 loss might correlate with human pancreatic cancer prognosis.

The detailed comments are as follows:

Major:

1. In Fig. 2c, the KO efficiency is not good enough.
2. In Fig. 2h,i, the representative images here could not support the conclusion. Need to repeat the experiments with improved conditions.
3. In Fig. 2, what are the potential mechanisms causing the inconsistent mutual effects between AP2S1 and AP2M1 depletions? e.g. genetic background? It is not clear to what extent of AP2S1 and AP2M1 contribute to cell growth reduction effects here because of the insufficient KO.
4. In Fig. 3, it looks like the AP2S1-depleted cells are resistant to DFO treatment in iron uptake. It is not clear that the resistance to the effects of DFO on cell growth is due to compensatory mechanisms in iron uptake or cell survival program.
5. In Fig. 5e, it looks like there was significant leakage of sgAP2S1 expression in some tumors, e.g. tumors #4 and #5. The AP2S1 expression looks even higher after Dox induction. It is not clear why the authors observe a consistent increase in the PM-bound ITGB1 here.
6. On the other hand, it is not clear why AP2S1 KO significantly reduces total ITGB1 expression. The authors should explain.
7. Why are the cells unable to activate the transferrin-independent pathways for iron uptake in vitro? The authors should attempt to address the question or at least discuss it.
8. The authors propose that the downregulation of PTEN is the potential mechanism underlying the enhanced tumor growth with AP2 loss. They should demonstrate if that alters the downstream signaling to activate the survival/proliferation gene program.

Minor:

1. The authors need to show if the sgRNAs also affect the expressions of the other genes, like Fig. 2b, c.
2. It is not clear if the figure here is correct. If it means Fig. 2b, c, there is no cell growth data. If it means Fig. 2d, the data demonstrated impaired cell growth.
3. The discussion largely repeats the results, the authors should discuss the potential compensatory mechanisms and the difference between in vitro and in vivo microenvironments (e.g. hypoxia, ECM complexity, nutrient levels, etc.)

Reviewer #2

(Remarks to the Author)

Through a targeted sgRNA library screening, the authors identified adaptor protein complex-2 (AP2) is essential for pancreatic cancer cell growth in vitro but suppresses pancreatic tumor growth in vivo. Based on RNA-seq analysis, they proposed that AP2 is required for transferrin receptor endocytosis in vitro, which is responsible for transferrin iron uptake. However, alternative iron import pathways are activated in tumors with AP2 loss. Based on plasma membrane proteomics analysis, they proposed that integrins retained on the membrane reduces the expression of the tumor suppressor PTEN. While the study is interesting, there are a few concerns:

1. "Perhaps then AP2 cargos on the cell surface may provide an accessible way to stabilize PTEN to inhibit tumorigenesis." PTEN seems to behave the similar way as AP2. Can this contrast phenotype between in vitro and in vivo solely due to the impact on PTEN but not the iron?
2. The authors reported "Second, TFRC exhibited higher expression in culture, while the genes SLC11A1, MCOLN1, SLC40A1, SLC22A17, and SCARA5, involved in alternative iron-internalization pathways, were more highly expressed in tumors, arguing for a shift from TFR1-dependent to -independent iron import from culture to tumors." Although the authors cited reference papers to support the important functions of these genes, they failed to identify which one was responsible for this manuscript. By the way, SLC40A1 is an iron export protein. Thus, its upregulation does not necessarily support the hypothesis.
3. Figure 1, the results from CFPAC-1 and HPAF-II do not seem to support the central hypothesis. There is no discussion in the manuscript.
4. Figure 2, KO efficiency was poor in most experiments, especially in 2C and 2L.
5. Figure 3B, it is hard to interpret the results. Quantification and better illustration are needed. The color for 3E mitoFerro Green is weird. In Figure 3F, the basal level of mitochondrial iron levels and the effect of DFO is not different between shCTRL and shAP2M1. "Perhaps then, targeting AP2 may provide a way to sensitize cancers to iron chelators, which to date have proven ineffective in clinical cancer trials." This statement lacks experimental support and seems to be contradictory to figure 3G.
6. Figure 5E, pFAK needs to be quantified.
7. Figure 5A, the variation is large compared to Figure 6C. Any explanation?
8. In addition, the rebuttal didn't address the critical comments experimentally and didn't incorporate replies in the main text.

Reviewer #3

(Remarks to the Author)

In this manuscript, Zimmerman and co-authors describe the differential effect of the down regulation of the AP2 complex in pancreatic cancer cell growth in vitro and in vivo. While the depletion of AP2 strongly prevents cell proliferation in vitro, it results in increased growth in tumours. The authors began to look at the mechanisms behind this, and concluded that the lack of growth in 2D upon AP2 loss was due to defects in iron uptake, while the increased proliferation in tumours was due to inhibition of integrin endocytosis, resulting in sustained adhesion signalling. The findings are novel and intriguing, however I believe more evidence needs to be provided to substantiate the conclusions.

1. The tumour promoting effect of AP2 loss can be observed in tumour spheroids in vitro. How is iron import regulated in this context? Did the authors observed changes in Transferrin receptor and the other transporters that are changed in vivo? If so, what promotes the switch from transferrin to other mechanisms of iron uptake? How does this observation fits with their focus on microenvironmental control?
2. A key difference between 2D/3D and in vitro/in vivo is substrate rigidity. It would be beneficial to assess the role of AP2 in controlling cell growth on substrates of different stiffness.
3. The different cell lines tested show a differential response to AP2 deletion - why is that? Is this related to different driver mutations? Are there differences in AP2 expression across the different cell lines? Do they have differences in transferrin expression? Is integrin trafficking regulated by different pathways?
4. The authors mainly used plasma membrane proteomics to demonstrate changes in integrin trafficking. I believe this has to be substantiated with more direct measurements of integrin endocytosis. Imaging-based approaches are suitable to use in 3D spheroids. In addition, the authors should also measure the total levels of all the integrins in the cells, and not only the surface levels. From the blot in figure S5c, it looks like there are big changes in some integrin levels in the whole-cell lysate,

not just at the plasma membrane. This could be a consequence of reduced endocytosis and consequent degradation, but the authors need to demonstrate that this is indeed the case.

5. Several pathways have been described to control integrin endocytosis, not just AP2. There are many contexts where integrin internalisation is clathrin-independent, therefore it is essential that the authors determine that the endocytosis of the integrin isoforms identified is indeed Clathrin and AP2 dependent in these cell lines

6. Similarly, the fact that FAK phosphorylation is affected by AP2 deletion doesn't necessarily mean that adhesion signalling is required for cell growth by AP2 loss. The authors should demonstrate this by inhibiting integrins (Several isoform-specific blocking antibodies are well characterised) and adhesion signalling and check the effects on AP2 dependent cell growth.

Reviewer #4

(Remarks to the Author)

The novelty of the study is the identification of two opposing tumorigenesis related mechanisms that act in vitro and in vivo regarding AP2 complex and endocytosis. The CRISPR screen was performed in a rather big panel of cell lines both in vivo and in vitro and experimentally validated, overall the opposite phenotypes upon AP2 related gene knockdown are adequately supported by the study. However, in vitro and especially in vivo mechanisms need further investigation. Meanwhile, the authors declare that endocytosis of pro-tumor proteins on the cell surface that activate PTEN is a novel mechanism leading to tumorigenesis inhibition, which requires further testing. Meanwhile, the clinical relevance of such mechanism would also require direct in vivo testing.

1. The in vitro and in vivo phenotypes seem distinct. Their relationship is not clear. The discussion of why one mechanism works in one situation needs accompany of why the other mechanism would not be dominant under the same circumstance. For example, TRF1 endocytosis inhibition by AP2 loss in vitro leads to reduced cell growth through iron intake. Would the pro-tumor receptor clustering be relevant in vitro? If plating cells on collagen or FN to induce integrin clustering, would iron phenotype be reduced? 3D growth and in vivo experiments showed the growth phenotype under shAP2M1, how about the role of receptors, which receptor is critical, would overexpression of such receptor overcome the growth phenotype?

The in vitro phenotype could represent how cells survive in low adhesion situations in vivo, such as in circulation, or in in vivo population that expresses less adhesion receptors, which could be bioinformatically analyzed using sc-RNAseq data and FACs sorting followed by qPCR. How about sc-RNAseq data from CTCs? In summary, the two separate functions of the AP2 complex may both be important in vivo, just depending on contexts, more evidence should be provided to delineate the mechanisms in clinically relevant processes.

2. Bioinformatics analysis should be done more specifically on tumor cells. TCGA analysis come from bulk RNAseq data. The gene expression reflects all cells in the TME but not just tumor cells. To distinguish cell type-specific expression profiles, tools such as CYBERSORTx could be used prior to survival analysis.

3. Protein level correlation need to be tested in a large panel of patients. Correlation between key receptor(s) localization on the cell surface, AP2 complex expression and overall survival need to be validated in clinical samples to supplement the gene expression data from TCGA, because 1. as mentioned previously tumor cell expression but not stromal expression is related and 2. the proposed protein endocytosis leads to protein level changes but not necessarily transcriptional changes (Fig 6F-H assumes that there are transcriptional changes of the related endocytosed receptors, which could be true but direct testing would be necessary).

4. Under current model, drugs that block clathrin-mediated endocytosis would inhibit tumor growth in vivo, while tumors knocked down of key integrins (the key receptor among the 17 overlapping receptors between BxPC3 and PANC1) would not respond to such drug. This is a key testing that is necessary to support the proposed in vivo mechanism.

Version 2:

Reviewer comments:

Reviewer #1

(Remarks to the Author)

The authors addressed the reviewer's comments with significant works, including new experiments and discussions. The reviewer appreciates their efforts. The revised manuscript demonstrated the novel roles of AP2-loss in cancer biology, revealing the crosstalk between membrane trafficking and signaling that regulates tumor progression.

Reviewer #3

(Remarks to the Author)

All my concerns have now been addressed and I believe the manuscript is significantly stronger and can now be accepted for publication.

I think it would be better to specify in the manuscript the actual targets of cilengitide, rather than simply saying integrin inhibitor.

Reviewer #4

(Remarks to the Author)

1. The revised version increased adequately the mechanistic evidence delineating the in vivo and in vitro differences. Discussion session now mostly discusses caveats, a clear summary of the 2D/3D/in vivo differences should be present in the discussion.

2. Figure S5h,i, as indicated in the response to comment 1, are not there in Fig.S5, nor is the text.

Reviewer #5

(Remarks to the Author)

The authors have addressed most of the concerns raised in the previous round of review. However, the concern regarding the TfR1-independent mechanism that supplies iron for cancer cell growth in in vivo or 3D culture has not been convincingly addressed. As the difference in iron transport between the in vitro and in vivo systems in response to AP-2 deficiency is the key scientific question of the study, stronger evidence is needed to demonstrate whether or not, in in vivo, tumor cell iron content is altered in the loss of AP-2. Decreased expression of IREB2 (Figure S3F) only indicates there might be a higher iron content in 3D versus 2D culture. In fact, the unaltered expression of IREB2 in 2D cultured cells between the WT and AP-2 knockout does not support intracellular iron deficiency in AP-2 knockout cells (due to impaired TfR1 endocytosis), because iron deficiency potentially leads to upregulation of IREB2 protein expression. The marked increase in SLC40A1 (Figure S3G), which encodes iron exporter ferroportin, would suggest iron deficiency in 3D cultured tumor cells. Yet, it remains possible that the upregulation of SLC40A1 is a feedback mechanism to prevent intracellular iron overload. Therefore, it becomes important to show the differences in intracellular iron content between 2D and 3D/tumor tissue with and without the knockout of AP-2. A few markers are suggested in the following. Ferritin (heavy chain or light chain), an iron binding protein that is widely used as a marker of intracellular iron status. FerroOrange staining of live cells that shows labile Fe²⁺ content. TfR1 protein expression should be also determined, although the focus of this study is on the effects of AP-2 knockout on TfR1 localization. The reason is that the expression of TfR1 is sensitive to changes in intracellular iron content.

Reviewer #1

Major Comment 1: The efficiency of sgRNAs at reducing AP2M1 and AP2S1 levels in CFPAC-1 cells appears to be poor in Figure 2c.

Reply: *We agree and as requested* we posited that CFPAC-1 cells in which AP2M1 or AP2S1 were efficiently knocked out were being lost from the population due to reduced iron internalization, hence the immunoblot only reported on those cells still expressing these proteins. To overcome this negative selection, we repeated the analysis, but cultured CFPAC-1 cells the presence of ammonium ferric citrate as a source of transferrin-independent iron during Cas9-mediated gene inactivation of *AP2M1* and *AP2S1* in triplicate independent cultures, followed by immunoblot of each triplicate sample. This method led to a more significant reduction in AP2M1 and AP2S1 protein levels. We thus replaced the previous Figure 2c with this revised experiment and updated the methods and text to reflect this change.

Major Comment 2: The poor knockdown of AP2M1 and AP2S1 in CFPAC-1 cells is at odds with the reduced proliferation of these cells reported in Figure 2h,i.

Reply: *We agree and as requested*, we retested three times the proliferation of three triplicate cultures of CFPAC-1 cells that we validated to have reduced AP2M1 or AP2S1 protein expression (see major comment #1), again finding reduced proliferation. We thus replaced the previous Figure 2 h,i with this revised experiment and updated both the methods and text to reflect this change.

Major Comment 3: In Figure 2, what are the potential mechanisms causing the inconsistent mutual effects between AP2S1 and AP2M1 depletions? e.g. genetic background? It is not clear to what extent of AP2S1 and AP2M1 contribute to cell growth reduction effects here because of the insufficient knockdown of these genes.

Reply: *We agree and as requested*, find that when CFPAC-1 cells are cultured with a source of transferrin-independent iron, the levels of AP2S1 and AP2M1 are indeed reduced by these guides (see major comment #1), which is consistent with the reduced proliferation of the resultant cells (see major comment #2).

Major Comment 4: It appears that AP2S1-depleted cells are resistant to DFO treatment in iron uptake in Figure 3. It is not clear that the resistance to the effects of DFO on cell growth is due to compensatory mechanisms in iron uptake or cell survival program.

Reply: *We revised the text to clarify* that the purpose of this experiment was to evaluate whether cells lacking AP2 were already depleted of iron and consequently would be resistant to further chelation by DFO, which was observed. However, as this is a minor confirmatory experiment we moved these data to the supplement.

Major Comment 5: In Figure 5e, it appears that there is significant leakage of sgAP2S1 expression in some tumors, e.g. tumors #4 and #5. The AP2S1 expression looks even higher after Dox induction. It is not clear why the authors observe a consistent increase in the PM-bound ITGB1 here.

Reply: As requested, we note here that the low levels of AP2S1 protein in these two tumors was likely a product of tumor-to-tumor variation (Figure 5e), as it is quite challenging to detect this protein, which we now note in the text.

Major Comment 6: Explain why reducing AP2S1 expression significantly reduces total ITGB1 levels in Figure 5e.

Reply: As requested, we speculate in the discussion that by altering the endocytosis rate of integrins overall trafficking, including to the lysosome for degradation, is also altered (*also see reviewer 3 comment # 4*).

Major Comment 7: The authors should attempt to address or at least discuss why cells are unable to activate the transferrin-independent pathways for iron uptake in vitro.

Reply: As requested, based on a suggestion from reviewer 3 (*see their comment #3*), we explored whether substrate stiffness might explain the changes in iron metabolism in vitro versus in vivo. We thus grew BXPC-3 cells on different substrate stiffnesses with and without reducing AP2M1, but found no effect on the expression of genes involved in iron metabolism. We thus explored another hypothesis. Spheroid and tumor growth have been implicated in the induction and stabilization of HIF1 α , and there is a known connection between iron concentration in cells and HIF transcription factors. However, we observed minimal changes in HIF1 α expression in BXPC-3 cells grown in 3D versus 2D culture. We thus speculate in the discussion why cells are unable to activate the transferrin-independent pathways in vitro in the revised text.

Major Comment 8: The authors propose that the downregulation of PTEN is the potential mechanism underlying the enhanced tumor growth with AP2 loss. They should demonstrate if that alters the downstream signaling to activate the survival/proliferation gene program.

Reply: As requested, we devoted significant effort towards this question, but failed to connect AP2 loss to a decrease in PTEN outside of tumors derived from BXPC-3 cells (*also see our reply to comment #1 from reviewer 2*). Since the most reproducible change associated with the enhanced tumor growth of cells lacking AP2 was increased integrin signaling, we instead revised the text to focus on integrins as the likely source of the tumor suppressive activity of AP2 in vivo.

Minor Comment 1: Confirm that sgRNAs also affect the expressions of the other genes in Figure 2b,c.

Reply: As requested, we added an immunoblot demonstrating that Cas9-mediated knockout of AP2S1 reduced AP2M1 levels in BXPC-3 cells and vice versa. We include these new data as Figure S2b and updated the methods and text to reflect this change.

Minor Comment 2: There appears to be a reference to a figure that is unclear. If it means Figure 2b,c there is no cell growth data. If it means Figure 2d, the data demonstrated impaired cell growth.

Reply: *Thank you* for catching this, we have revised the text accordingly.

Minor Comment 3: The discussion largely repeats the results, the authors should discuss the potential compensatory mechanisms and the difference between in vitro and in vivo microenvironments (e.g. hypoxia, ECM complexity, nutrient levels, etc.)

Reply: *As requested*, we revised the text accordingly.

Reviewer #2

Comment 1: PTEN seems to behave the similar way as AP2. Can this contrast phenotype between in vitro and in vivo solely be due to the impact on PTEN but not the iron?

Reply: As requested, we explored this avenue further. However, we did not observe a decrease in PTEN upon targeting AP2 in BXPC-3 cells in culture. Additionally, we could rescue the loss of BXPC-3 (and also CFPAC-1) cells growth in culture upon disrupting AP2 with the addition of transferrin-independent iron in the form of ammonium ferric citrate, yet PTEN levels were again unaffected. The decrease in PTEN appears to be specific to BXPC-3 cells when grown as tumors. Since the most reproducible change associated with the enhanced tumor growth of cells lacking AP2 was increased integrin signaling, we instead revised the text to focus on integrins as the likely source of the tumor suppressive activity of AP2 in vivo. We appreciate the suggestion to dig into PTEN further, as these new experiments strengthened the manuscript by focusing our efforts on integrins instead of PTEN.

Comment 2: The authors reported that "...TFRC exhibited higher expression in culture, while the genes SLC11A1, MCOLN1, SLC40A1, SLC22A17, and SCARA5, involved in alternative iron-internalization pathways, were more highly expressed in tumors, arguing for a shift from TFR1-dependent to -independent iron import from culture to tumors." Although the authors cited reference papers to support the important functions of these genes, they failed to identify which one was responsible for this manuscript. By the way, SLC40A1 is an iron export protein. Thus, its upregulation does not necessarily support the hypothesis.

Reply: As requested, due to the challenge of testing every one of these genes in vivo and the distinct possibility that they may all contribute to restoring iron import, we instead asked whether there is evidence of tumors having restored iron import, the end product of all these pathways. We thus used the *Iron Responsive Element Binding protein 2* (IREB2), as a measure of iron status in cells, as this protein is upregulated to stabilize iron metabolism transcripts in conditions of iron deprivation. We find from total transcriptomics that *IREB2* (and also the paralog *AC01*) mRNA levels are higher in cultured cells ($n=3$) compared to tumors ($n=5$), consistent with low iron and corresponding the lethality of AP2 loss in vitro. By immunoblot, we show that IREB2 protein is elevated in duplicate cultures of BXPC-3 cells when cultured in 2D versus 3D. We include these new data as Figure S3f-h, updated the methods to reflect this change, and revised the discussion to note that these data support iron import being restored in vivo, coincidental with an increase in the genes involved in alternative iron-internalization pathways. Lastly, we appreciate pointing out that SLC40A1 is an iron export protein. We have updated the text to reflect that we observed changes in iron transport rather than a strict increase in transferrin-independent iron uptake pathways. We included SLC40A1 in our analysis as inhibition of this solute carrier decreases overall iron internalization, likely through an interaction and regulation of the DMT1 metal ion transporter. This reference has been added to the manuscript as well.

Comment 3: The results from CFPAC-1 and HPAF-II in Figure 1 do not seem to support the central hypothesis.

Reply: As requested, we agree with the reviewer that AP2 loss did **NOT** enhance tumor growth of CFPAC-1 and HPAF-II cells. Instead of this being seen as not supporting the central hypothesis, we took advantage of this observation to compare cells that were (PANC-1, MIAPACA-2, and BXPC-3) versus were not (CFPAC-1 and HPAF-II) more tumorigenic when AP2 was disrupted for mechanistic studies. The premise being that any effect of AP2 loss in tumors

derived from PANC-1, MIAPACA-2, and BXPC-3 cells should ***NOT*** be observed in tumors derived from CFPAC-1 and HPAF-II cells. Ultimately, this comparison revealed that while the loss of AP2 reprogrammed the plasma membrane proteome in both CFPAC-1 and BXPC-3 cells, consistent with AP2-dependent endocytosis being disrupted in both, integrins were trapped on the cell surface of BXPC-3, but not CFPAC-1 derived tumors, with transcriptomics further supporting that this was associated with a pro-tumorigenic signature in BXPC-3, but not CFPAC-1 cells. We have since expanded these studies to now show that while AP2 promotes endocytosis of the transferrin receptor in both BXPC-3 and CFPAC-1 cells, it only enhances ITGB1 (integrin beta-1) endocytosis in BXPC-3 cells, suggesting AP2 discriminates cargo proteins in a cell type dependent manner (see comment 4 below). This topic has been added to the discussion.

Comment 4: The knockout efficiency was poor in most experiments in Figure 2, especially for CFPAC-1 cells.

Reply: *We agree and as requested* we posited that CFPAC-1 cells in which AP2M1 or AP2S1 were efficiently knocked out were being lost from the population due to reduced iron internalization, hence the immunoblot only reported on those cells still expressing these proteins. To overcome this negative selection, we repeated the analysis, but cultured CFPAC-1 cells the presence of ammonium ferric citrate as a source of transferrin-independent iron during Cas9-mediated gene inactivation of AP2M1 and AP2S1 in triplicate independent cultures, followed by immunoblot of each triplicate sample. This method led to a more significant reduction in AP2M1 and AP2S1 expression. We thus replaced the previous Figure 2c with this revised experiment and updated the methods and text to reflect this change.

Comment 5: Quantification and better illustration are needed for Fig 3B. The color for 3E mitoFerro Green is weird. In Fig 3F, the basal level of mitochondrial iron levels and the effect of DFO is not different between shCTRL and shAP2M1. The statement “Perhaps then, targeting AP2 may provide a way to sensitize cancers to iron chelators, which to date have proven ineffective in clinical cancer trials.” thus lacks experimental support and seems to be contradictory to Figure 3G.

Reply: *As requested*, we repeated Figure 3B with BXPC-3, but now also CFPAC-1 cells. Specifically, we quantified transferrin endocytosis with fluorescent-labelled transferrin in 10 images from duplicate cultures of both cell lines, and included the Pitstop2 to block clathrin-dependent endocytosis based on a suggestion from reviewer 3 for a related experiment (see their comment #5). These new improved images and now quantification are shown as Figures 3b-e, and we updated the methods and text to reflect these new data.

As requested, we repeated the experiment shown in Figure 3e to obtain better images that are depicted in a black and white to address the issue of mitoFerro Green coloring.

As requested, to address this point we depleted iron levels with the iron chelator DFO in duplicate cultures of BXPC-3 cells, and then continued treatment with DFO, transferrin to provide transferrin-based iron, or ammonium ferric citrate to provide transferrin-independent iron, and measured the amount of cellular iron using mitoFerro Green (10 images per condition). We find that there is no difference in mitoFerro Green levels in the absence or presence of AP2 when treated with DFO or ammonium ferric citrate. However, we do see reduced mitoFerro Green levels in cells treated with transferrin in the absence of AP2. We include these new data as Figure 3j,k, updated the methods to reflect this change, and revised the text to note that these data support AP2 specifically regulates transferrin-based iron import.

Finally, ***as requested*** the comment regarding AP2 sensitizing cells to iron chelators has been removed from the text, figure 3G has been moved to the supplement as it is a minor piece of evidence, and the text has been appropriately revised.

Comment 6: Figure 5E, pFAK needs to be quantified.

Reply: ***As requested***, we include this quantification in the revised Figure 5e,f.

Comment 7: Figure 5A, the variation is large compared to Figure 6C. Any explanation?

Reply: ***As requested***, Figure 6 has been deleted based on a suggestion from reviewer 4 (see their comment # 2).

Comment 8: In addition, the rebuttal didn't address the critical comments experimentally and didn't incorporate replies in the main text.

Reply: ***We apologize***, we did not realize we had the opportunity to make these adjustments at the stage of editorial review.

Reviewer #3

Comment 1: The tumour promoting effect of AP2 loss can be observed in tumour spheroids in vitro. How is iron import regulated in this context? Did the authors observed changes in Transferrin receptor and the other transporters that are changed in vivo? If so, what promotes the switch from transferrin to other mechanisms of iron uptake? How does this observation fits with their focus on microenvironmental control?

Reply: As requested, we performed qRT-PCR on the panel of six transcripts encoding proteins involved in iron metabolism, including the transferrin receptor, in two independent cultures of BXPC-3 versus CPPAC-1 assayed in triplicate when grown in 3D versus 2D cultures. We find that as in the case of tumors, these genes were upregulated in 3D versus 2D culture. We include these new data as Figure S3g,h, updated the methods to reflect this change, and revised the text to note that these data support a transition in iron transport from 2D to 3D culture akin to what was observed in tumors.

As requested, while the exact mechanism underlying this switch is unclear (see our reply to comment #2 from reviewer 2), we addressed whether tumors have more iron, the end product of all these pathways. We thus used the *Iron Responsive Element Binding protein 2* (IREB2) as a measure of iron status in cells, as this protein is upregulated to stabilize iron metabolism transcripts in conditions of iron deprivation. We find from total transcriptomics that *IREB2* (and also the paralog *AC01*) mRNA levels are higher in cultured cells ($n=3$) compared to tumors ($n=5$), consistent with low iron and corresponding the lethality of AP2 loss in vitro. By immunoblot, we show that IREB2 protein is elevated in duplicate cultures of BXPC-3 cells when cultured in 2D versus 3D. We include these new data as Figure S3f-h, updated the methods to reflect this change, and revised the discussion to note that these data support iron import being restored in vivo, coincidental with an increase in the genes involved in alternative iron-internalization pathways. We also discuss these observations within the context of the in vivo environment.

Comment 2: A key difference between 2D/3D and in vitro/in vivo is substrate rigidity. It would be beneficial to assess the role of AP2 in controlling cell growth on substrates of different stiffness.

Reply: As requested, we thought this was a great idea and assayed the effect of AP2 loss on the growth of BXPC-3 and CFPAC-1 cells in 96-well Matrigen plates coated with acrylamide gels of varying stiffnesses. We found the only statistically significant difference was in uncoated wells in which AP2-depleted cells grew more slowly than control cells, as expected based on the poor growth of these cells in tissue culture. All cultures grew less well on the acrylamide gels compared to same cells cultured on plastic lacking AP2, and the presence or absence of AP2 did not change their growth. Furthermore, qPCR of iron metabolism transcripts on different substrate stiffnesses did not reveal any changes in these genes. While this line of investigation was not fruitful, we nevertheless discuss potential features of 3D culture that restore iron import upon AP2 loss (see our reply to major comment # 7 from reviewer 1).

Comment 3: The different cell lines tested show a differential response to AP2 deletion - why is that? Is this related to different driver mutations? Are there differences in AP2 expression across the different cell lines? Do they have differences in transferrin expression? Is integrin trafficking regulated by different pathways?

Reply: As requested, we thought this was another great idea and favor the last model proposed by the reviewer. In more detail, we performed plasma membrane proteomics in the original submission in the absence and presence of AP2 in tumors derived from 1) CFPAC-1 cells, in

which AP2 loss has no effect on tumor growth, 2) PANC-1 cells, which grow better in vivo upon AP2 loss, and 3) BXPC-3 cells, in which AP2 loss enhances tumor growth the most of all cell lines tested. We found that AP2 loss in tumors derived from all three of these cell lines resulted in significant changes in the plasma membrane proteome, confirming that the loss of AP2 does indeed affect AP2-dependent endocytosis. However, the changes in the plasma membrane proteome of CFPAC-1-derived tumors did **NOT** overlap with tumors derived from BXPC-3 or PANC-1 cells (see Figure 5a-d). This led us to investigate the commonalities in BXPC-3 and PANC-1 cells, which revealed a reduction in integrin endocytosis, which at least partially explains the difference between tumors with enhanced growth and those without (see Figure 5b-n). We have since expanded these studies to show that while AP2 enhances transferrin endocytosis in both BXPC-3 and CFPAC-1 cells, it only enhances Integrin beta-1 endocytosis in BXPC-3 cells (see comment #4 below for a description of this experiment). We include these new data as Figure 4g-j, updated the methods to reflect this change, and revised the discussion to note that these data support AP2 discriminates cargo proteins in a cell-type dependent manner.

Comment 4: The authors mainly used plasma membrane proteomics to demonstrate changes in integrin trafficking. I believe this has to be substantiated with more direct measurements of integrin endocytosis. Imaging-based approaches are suitable to use in 3D spheroids. In addition, the authors should also measure the total levels of all the integrins in the cells, and not only the surface levels. From the blot in Figure S5c, it looks like there are big changes in some integrin levels in the whole-cell lysate, not just at the plasma membrane. This could be a consequence of reduced endocytosis and consequent degradation, but the authors need to demonstrate that this is indeed the case.

Reply: *As requested*, we quantified ITGB1 (Integrin beta 1) endocytosis with a fluorescent-labelled anti-ITGB1 antibody from 10 images from duplicate cultures of BXPC-3 and CFPAC-1 cell lines, and included the Pitstop2 to block clathrin-dependent endocytosis based on a suggestion from reviewer 3 for a related experiment (see their comment #5). These new improved images and now quantifications are shown as Figures 4g-j, we updated the methods, and revised the text to note that these data are consistent with BXPC-3, but not CFPAC-1 cells regulating integrin endocytosis in an AP2-dependent manner.

As requested, we note that total levels of ITGA2 and ITGB6 were increased in five tested tumors from BXPC-3 cells upon dox-induction of *AP2M1* sgRNA compared to five control tumors that were not induced with doxycycline (see Figures 4e and S4a).

As requested, we explored the reviewer's model, and demonstrate that despite the noted high levels in the total amount of ITGA2 and ITGB6 protein upon AP2 loss in vivo, analysis of the transcriptomes of the tumors derived from BXPC-3 cells with or without AP2 revealed that the mRNA corresponding to these genes was not increased in the absence of AP2 (Figure 5b). We now note these data in the text, and speculate in the discussion that by altering the endocytosis of integrins, overall trafficking - including to the lysosome for degradation- is also altered.

Comment 5: Several pathways have been described to control integrin endocytosis, not just AP2. There are many contexts where integrin internalization is clathrin-independent, therefore it is essential that the authors determine that the endocytosis of the integrin isoforms identified is indeed Clathrin and AP2 dependent in these cell lines.

Reply: *As requested*, we assayed for AP2- and clathrin-mediated endocytosis dependent internalization of ITGB1 in BXPC-3 and CFPAC-1 cells (see comment #4 above for a description of this experiment). Indeed, we observed ITGB1 was internalized in an AP2- and clathrin-independent manner in CFPAC-1 cells but in an AP2- and clathrin-dependent manner in BXPC-

3 cells (Figure 4g-j). We believe this explains the growth phenotype differences in CFPAC-1 and BXPC-3 cells and also provides evidence that AP2 has the ability to discriminate cargoes since transferrin internalization was AP2-dependent in both cell types. Thank you for this suggestion, as it has pointed towards a potential mechanism.

Comment 6: Similarly, the fact that FAK phosphorylation is affected by AP2 deletion doesn't necessarily mean that adhesion signaling is required for cell growth by AP2 loss. The authors should demonstrate this by inhibiting integrins (Several isoform-specific blocking antibodies are well characterized) and adhesion signaling and check the effects on AP2 dependent cell growth.

Reply: As requested, BXPC-3 cells cultured in quadruplicate as tumor spheroids in the absence and presence of AP2 were immunoblotted for total and phosphorylated FAK. We observe that an increase in phosphorylated FAK in the absence of AP2. We then repeated the analysis in triplicate, but in the presence of the integrin inhibitor cilengitide or the FAK inhibitor PF-573228, and find a decrease of FAK phosphorylation back to base level upon treatment with either compound. We have included these new data as Figure 4k-m. In the same manner we have shown that AP2 depletion does not alter FAK phosphorylation in CFPAC-1 cells when grown in 3D culture (Figure S4g).

As requested, to test if these inhibitors also altered BXPC-3 cell growth in 3D culture, lentiviral constructs encoding dox-inducible GFP or Red Fluorescent Protein (RFP) and Control shRNA (*shCTRL*) or *shAP2M1* were used to generate GFP-*shCTRL*, RFP-*shCTRL*, and RFP-*shAP2M1* BXPC-3 cells for growth competition assays. BXPC-3 cells expressing RFP-*shCTRL* or RFP-*shAP2M1* were mixed with an equal number of GFP-*shCTRL* cells, grown in 3D culture, and treated with a vehicle control, PF-573228, or cilengitide. For each condition six replicate cultures were tested in two independent experiments. Seven days later the number of GFP- and RFP-positive cells were counted and the ratio of the two determined. This revealed that *shAP2M1*-expressing BXPC-3 cells showed enhanced growth in 3D culture compared to *shCTRL*-expressing cells, an effect that was reduced to base when cultured in either inhibitor. These new data are presented in Figure 4n. In the same manner, we shown that AP2 depletion does not alter CFPAC-1 cell growth in 3D culture (Figure 2s). We have updated the methods and results to reflect these new experiments, and revised the discussion to note that these results are consistent with the proposed mechanism that AP2 loss leads to enhanced tumor and spheroid growth through integrin mediated FAK phosphorylation.

Reviewer #4

Comment 1: The in vitro and in vivo phenotypes seem distinct. Their relationship is not clear. The discussion of why one mechanism works in one situation needs accompany of why the other mechanism would not be dominant under the same circumstance. For example, TRF1 endocytosis inhibition by AP2 loss in vitro leads to reduced cell growth through iron intake. Would the pro-tumor receptor clustering be relevant in vitro? If plating cells on collagen or FN to induce integrin clustering, would iron phenotype be reduced? 3D growth and in vivo experiments showed the growth phenotype under shAP2M1, how about the role of receptors, which receptor is critical, would overexpression of such receptor overcome the growth phenotype?

Reply: *As requested*, we tested the prediction that the loss of proliferation in culture due to reduced endocytosis of the transferrin receptor upon AP2 loss is dominant over the signaling generated by retaining integrins on the plasma membrane. We find that integrins are indeed retained in the plasma membrane in BXPC-3 cells upon loss of AP2 in culture (see comment 4 from reviewer 3 for experimental details). Furthermore, two independent cultures of BXPC-3 and CFPAC-1 cells grown in 2D versus 3D in the absence and presence of AP2 were immunoblotted for total and phosphorylated FAK, which revealed high levels of phosphorylated FAK in 2D cultures of BXPC-3 cells, regardless of AP2 status, while the low levels of phosphorylated FAK were increased in 3D culture of these cells upon the loss of AP2. CFPAC-1 cells displayed high levels of phosphorylated FAK in 2D culture, but undetectable levels in 3D, regardless of AP2 status. We include these new data as Figure S5h,i, updated the methods to reflect this change, and revised the test to note that FAK activity is likely saturated in 2D comparison to 3D culture, and therefore the cells do not further respond to integrin retention on the plasma membrane. Finally, we demonstrate that the integrin inhibitor cilengitide reduced the 3D growth of BXPC-3 cells imparted by the loss of AP2 (see comment 6 from reviewer 3 for experimental details). We include these new data as Figure 4k-m, updated the methods to reflect this change, and revised the discussion to note that these results are consistent with the proposed mechanism that AP2 loss leads to enhanced tumor and spheroid growth through integrin mediated FAK phosphorylation.

Comment 2: Bioinformatics analysis should be done more specifically on tumor cells. TCGA analysis come from bulk RNAseq data. The gene expression reflects all cells in the TME but not just tumor cells. To distinguish cell type-specific expression profiles, tools such as CYBERSORTx could be used prior to survival analysis.

Reply: *We thank the reviewer for this critical suggestion*, especially in the context of pancreatic cancer where much of the samples from TCGA contain a large proportion of other cell types. As requested, we deconvoluted the TCGA data with cybersortX and other software packages as suggested, which altered the correlations and patterns originally noted. We have therefore removed these analyses. We appreciate the comment to dig into this analysis further, which strengthened the manuscript by focusing our efforts on the underlying mechanism.

Comment 3: Protein level correlation need to be tested in a large panel of patients. Correlation between key receptor(s) localization on the cell surface, AP2 complex expression and overall survival need to be validated in clinical samples to supplement the gene expression data from TCGA, because 1.as mentioned previously tumor cell expression but not stromal expression is related and 2. the proposed protein endocytosis leads to protein level changes but not necessarily

transcriptional changes (Fig 6F-H assumes that there are transcriptional changes of the related endocytosed receptors, which could be true but direct testing would be necessary).

Reply: As stated above, the bioinformatic analysis of TCGA data has been removed in light of the analysis suggested above.

Comment 4: Under the current model, drugs that block clathrin-mediated endocytosis would inhibit tumor growth in vivo, while tumors knocked down of key integrins (the key receptor among the 17 overlapping receptors between BxPC3 and PANC1) would not respond to such drug. This is a key testing that is necessary to support the proposed in vivo mechanism.

Reply: *This is a great suggestion.* While we have not performed this exact experiment, we tested and found that the FAK and the integrin inhibitor cilengitide inhibit the increased FAK phosphorylation and proliferation observed in 3D culture upon AP2 loss as described above in response to comment #1.

Reviewer #1

Major Comment 1: “The authors addressed the reviewer's comments with significant works, including new experiments and discussions. The reviewer appreciates their efforts. The revised manuscript demonstrated the novel roles of AP2-loss in cancer biology, revealing the crosstalk between membrane trafficking and signaling that regulates tumor progression.”

Reply: We thank this reviewer for their time and support.

Reviewer #2 (withdrawn)

Reviewer #3

Comment 1: “All my concerns have now been addressed and I believe the manuscript is significantly stronger and can now be accepted for publication. I think it would be better to specify in the manuscript the actual targets of cilengitide, rather than simply saying integrin inhibitor.”

Reply: We thank this reviewer for their time and support. *As requested*, we have added the specific integrin dimers that cilengitide has been shown to inhibit.

Reviewer #4

Comment 1: “The revised version increased adequately the mechanistic evidence delineating the *in vivo* and *in vitro* differences. Discussion session now mostly discusses caveats, a clear summary of the 2D/3D/*in vivo* differences should be present in the discussion.”

Reply: We thank this reviewer for their time and support. *As requested*, we have expanded the discussion to address differences in the growth conditions.

Comment 2: “Figure S5h,l, as indicated in the response to comment 1, are not there in Fig.S5, nor is the text.”

Reply: We apologize for this oversight.

Reviewer #5

Comment 1: “...therefore, it becomes important to show the differences in intracellular iron content between 2D and 3D/tumor tissue with and without the knockout of AP-2. A few markers are suggested in the following. Ferritin (heavy chain or light chain), an iron binding protein that is widely used as a marker of intracellular iron status. FerroOrange staining of live cells that shows labile Fe²⁺ content. TfR1 protein expression should be also determined, although the focus of this study is on the effects of AP-2 knockout on TfR1 localization. The reason is that the expression of TfR1 is sensitive to changes in intracellular iron content.”

Reply: We thank the reviewer for their time in reviewing the revision, and for their expertise in iron metabolism. *As requested*, we compared the expression of IREB2, TfR1(TFRC), and FTH1 in 2D and 3D culture, as well as in tumors with and without AP2 by immunoblot. These additional

experiments strengthen the previous findings that AP2 is not necessary for sufficient intracellular iron in 3D culture and tumors that. We also attempted to use FerroOrange to assess intracellular iron levels in 2D and 3D culture, but unfortunately this dye did not reliably penetrate the tumor spheroids and was therefore not included in this revision. We apologize for this shortcoming.